palaeontology/evolution

Sandownidae, Thalassochelydia, cranial anatomy, labyrinth, turtles, evolution

**Author for correspondence:**
Serjoscha W. Evers
e-mail: serjoscha.ever@googlemail.com

# A re-description of *Sandownia harrisi* (Testudinata: Sandownidae) from the Aptian of the Isle of Wight based on computed tomography scans

Serjoscha W. Evers and Walter G. Joyce

Departement of Geosciences, University of Fribourg, Fribourg, Switzerland

SWE, 0000-0002-2393-5621

Sandownidae is an enigmatic group of Cretaceous–Paleogene turtles with highly derived cranial anatomy. Although sandownid monophyly is not debated, relationships with other turtles remain unclear. Sandownids have been recovered in significantly different parts of the turtle tree: as stem-turtles, stem-cryptodires and stem-chelonioid sea turtles. Latest phylogenetic studies find sandownids as the sister-group of the Late Jurassic thalassochelydians and as stem-turtles. Here, we provide a detailed study of the cranial and mandibular anatomy of *Sandownia harrisi* from the Aptian of the Isle of Wight, based on high resolution computed tomography scanning of the holotype. Our results confirm a high number of anatomical similarities with thalassochelydians and particularly *Solnhofia parsonsi*, which is interpreted as an early member of the sandownid lineage. Sandownids + *Solnhofia* show many cranial modifications related to the secondary palate and a durophagous diet. *Sandownia* is additionally highly derived in features related to its arterial circulation and neuroanatomy, including the endosseous labyrinth. Our results imply rapid morphological evolution during the early history of sandownids. Sandownids likely evolved in central Europe from thalassochelydian ancestors during the Late Jurassic. The durophagous diet of sandownids possibly facilitated their survival of the Cretaceous/Paleogene mass extinction.

## 1. Introduction

The Early Cretaceous (Lower Aptian: *Deshayites forbesi* Zone [1]) fossil turtle *Sandownia harrisi* is known from a single specimen

**Figure 1.** Photographs of the cranium of the holotype specimen of *Sandownia harrisi* (MIWG 3480). (*a*) Dorsal view. (*b*) Ventral view. (*c*) Left lateral view. (*d*) Right lateral view. (*e*) Anterior view. (*f*) Posterior view. Scale bar, 20 mm.

from the Isle of Wight that was initially described by Meylan *et al.* [2]. The holotype includes a well-preserved cranium (figure 1) and a partial lower jaw (figure 2).

New fossils described since the description of *Sandownia harrisi* have shown that the peculiar skull morphology of this turtle is shared by a number of other fossil turtle species, in particular *Angolachelys mbaxi* [3] from the Late Turonian of Angola, *Leyvachelys cipadi* [4] from the Late Barremian or Aptian of Colombia, and *Brachyopsemys tingitana* [5] from the Danian of Morocco. Tong & Meylan [5] named the group comprising these four turtles Sandownidae, and Evers & Benson [6] provided a phylogenetic definition for the name.

*Sandownia harrisi* was originally identified as an aberrant representative of Trionychoidea [2], a no longer recognized group of turtles consisting of the clades Kinosternoidea and Trionychia [7]. All phylogenetic analyses including more than one sandownid have confirmed the monophyly of this group with regard to other turtles [2–4,6,8]. However, the phylogenetic position of sandownids within the turtle tree remains unclear, with some authors identifying them variably as stem-chelonioids (e.g. [4,5,9]), but others as stem-cryptodires [3], stem-turtles [6], stem-pleurodires [8], trionychians (e.g. [10,11]), protostegids [12], or among the group now termed Thalassochelydia, which includes plesiochelyids, eurysternids, and thalassemydids (e.g. [3,6,12]).

Sandownids span at least 60 million years of evolutionary history, and survived the mass extinction associated with the Cretaceous/Paleogene boundary. Sandownids further had a wide geographical distribution, even in the Early Cretaceous. This can possibly be explained by their ecology, as sandownids are universally interpreted as secondarily marine turtles, because their fossils have been recovered from shallow marine depositional environments, and because the presence of extensive secondary palates and large triturating surfaces are consistent with marine feeding adaptations [3,5,6]. Depending on their phylogenetic position, sandownids could either be informative about the early evolution of extant sea turtles (chelonioids), or provide independent evidence for secondary marine adaptation. The latter is an important macroevolutionary field of research, as ecological transitions provide important insights into the tempo and mode of morphological evolution (e.g. [8,13–15]).

In this paper, we provide a re-description of the cranial anatomy of *Sandownia harrisi* based on 3D models segmented from CT scans of the holotype cranium and mandible. We make specific comparative reference to other sandownids and taxa that have repeatedly been proposed to be closely related to sandownids, particularly the Late Jurassic *Solnhofia parsonsi*. In addition to describing the

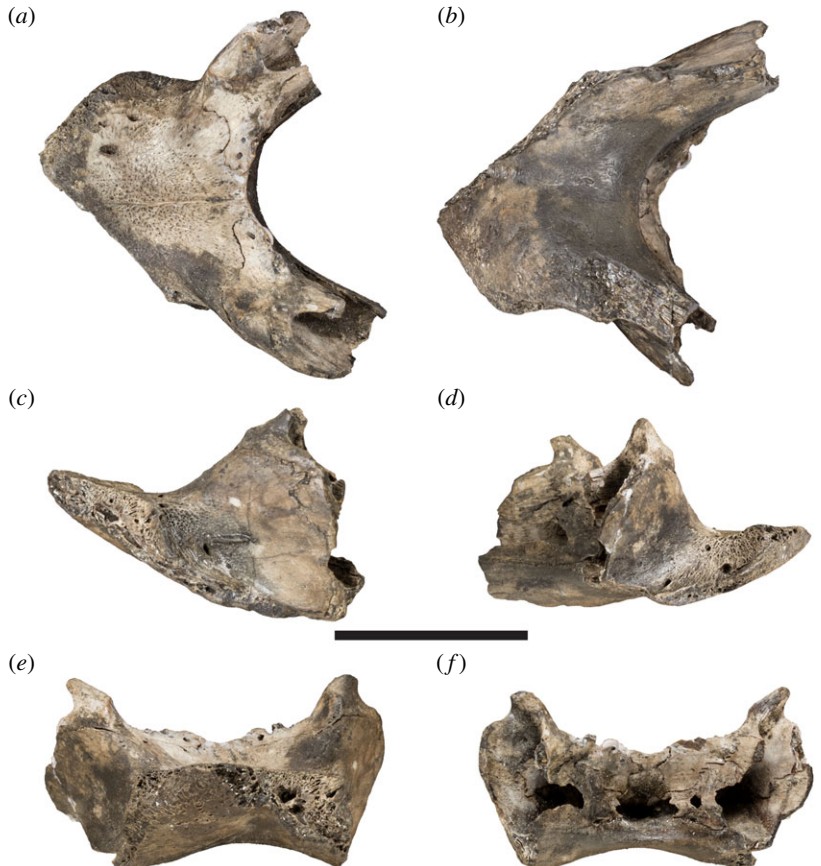

**Figure 2.** Photographs of the mandible of the holotype specimen of *Sandownia harrisi* (MIWG 3480). (*a*) Dorsal view. (*b*) Ventral view. (*c*) Left lateral view. (*d*) Right lateral view. (*e*) Anterior view. (*f*) Posterior view. Scale bar, 20 mm.

osteology, we also show the pattern of carotid circulation and the shape of the bony labyrinth based on digital endocasts of the respective cranial cavities.

## 2. Material and methods

High-resolution X-ray computed tomography (CT) scans were obtained for the cranium and mandible of the holotype specimen of *Sandownia harrisi*, MIWG 3480 at the University of Bristol, using a Metris X-Tek HMX ST 225 CT System. The cranium was scanned using a beam energy of 170 kV and a current of 180 µA; 1 ms exposure time; 1 frame per projection and 3141 projections; 0.5 mm Copper filter; resulting in a voxel size of 0.049549 mm. The mandible was scanned with the same parameters, except a current of 175 µA was used and 1500 projections were made. The voxel size (resolution) of the scan was 0.03053612017 mm. The resulting CT-scans were segmented in the software Mimics (v. 16.0–19.0; http://biomedical.materialise. com/mimics), and 3D models were exported as .ply files. Figures of digital renderings were compiled using the software Blender v. 2.71 (blender.org). CT-slice data as well as the 3D models are deposited at MorphoSource [16]. In the comparative description, we cite studies that describe the relevant taxa when the respective authors mention the features to which we refer. We use only the specimen numbers when referring to observations that we made, but which have not been reported previously.

## 3. Results

### 3.1. Description

#### 3.1.1. Nasal

Meylan *et al.* [2] initially reported nasals to be absent in *Sandownia harrisi*. However, we here identify in the CT data a small, median bone at the dorsal margin of the external naris as the fused nasals. The bone

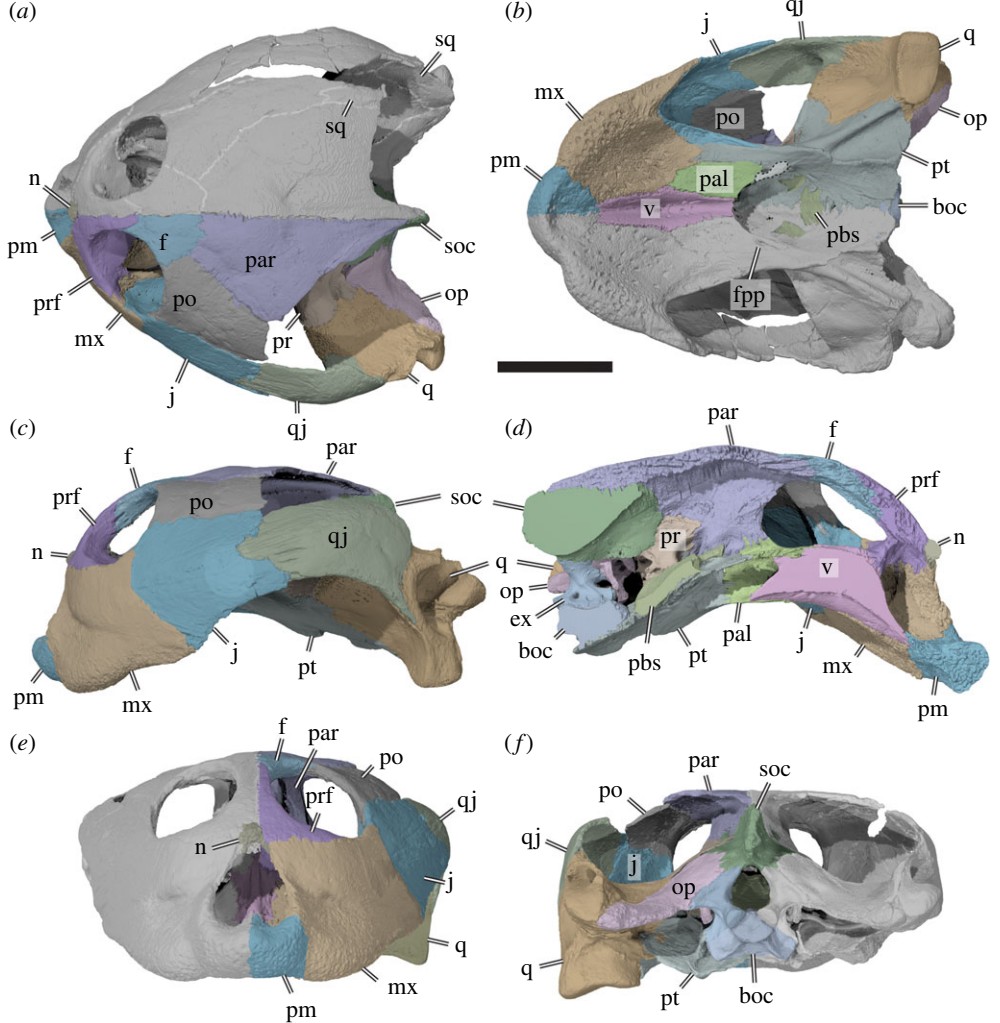

**Figure 3.** Three-dimensional renderings of the cranium of the holotype specimen of *Sandownia harrisi* (MIWG 3480). (*a*) Dorsal view. (*b*) Ventral view. (*c*) Left lateral view. (*d*) Medial view on left side of cranium. (*e*) Anterior view. (*f*) Posterior view. boc, basioccipital; ex, exoccipital; f, frontal; fpp, foramen palatinum posterius; mx, maxilla; j, jugal; n, nasal; op, opisthotic; pal, palatine; pbs, parabasisphenoid; par, parietal; pm, premaxilla; po, postorbital; pr, prootic; prf, prefrontal; pt, pterygoid; q, quadrate; qj, quadratojugal; soc, supraoccipital; sq, squamosal; v, vomer. Dotted line around part of the pterygoid in *b* indicates the small process that closes the foramen palatinum posterius. Suture traces have been highlighted in right cranial side of *a* to show additional features not preserved on the left side. Scale bar, 20 mm.

is damaged by weathering at its external surface, which could be why Meylan *et al.* [2] interpreted the nasals to be absent (figures 3*a,e* and 4*a*).

The CT scans do not show any internal sutures and the nasals are herein interpreted as fused. The fused nasals have a short ventrolateral process that extends along the maxilla within the margin of the external naris. Posteriorly, the nasals are contacted by the prefrontals.

Nasals are described to be present in the Texan material of *Leyvachelys cipadi* [17], but only indicated to be present in the figures of the Colombian material [4]. Mateus *et al.* [3] infer the presence of nasals for *Angolachelys mbaxi*, but report that they are missing in the preserved material.

### 3.1.2. Prefrontal

The prefrontal contacts the maxilla laterally, the frontal posteriorly, and the descending process contacts the palatine posteroventrally and the vomer ventrally, and forms the anterior margin of the enlarged foramen orbito-nasale ([2]; figure 3*a,c–e*). Additionally, the prefontals frame the fused nasals anteromedially. The prefrontal is excluded from the external naris (*contra* [2]), but its contribution to the orbit is extensive. Posteroventrally, the prefrontal expands to a mediolaterally broadened process that broadly overlaps the

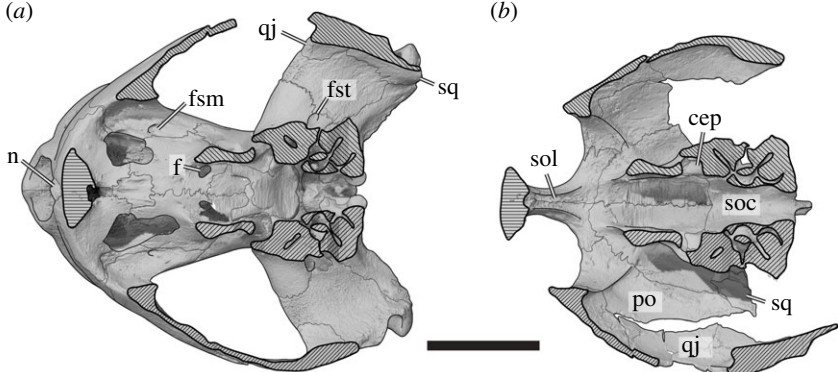

**Figure 4.** Three-dimensional renderings of the horizontally sectioned cranium of *Sandownia harrisi* (MIWG 3480) with overlaid interpretative line drawings. (*a*) Ventral portion of cranium in dorsal view. (*b*) Dorsal portion of cranium in ventral view. cep, cavum epiptericum; f, unnamed foramen in palatine; fsm, foramen supramaxillare; fst, foramen stapedio-temporale; n, nasal; po, postorbital; qj, quadratojugal; soc, supraoccipital; sol, sulcus olfactorius; sq, squamosal. Suture traces are thin lines. Hatched areas indicate section planes through the cranium. Scale bar, 20 mm.

maxilla within the orbital cavity. As a result, the prefrontal almost contacts the jugal and nearly excludes the maxilla from the orbital floor and the ventral margin of the orbit. As described by Meylan *et al.* [2], the interorbital bar, which is largely formed by the prefrontals, is very thin. The dorsal process of the prefrontal forming this bar is ventrally underlapped by the frontal, which extends anteriorly close to the fissura ethmoidalis. Although most of the crista cranii is formed by the frontal, this crest is anteriorly continuous on the ventral surface of the dorsal prefrontal process (figure 4*b*).

### 3.1.3. Frontal

The frontal of *Sandownia harrisi* is a relatively small bone, which anteriorly contributes to the thin interorbital bar where it contacts the prefrontal (figure 3*a*,*c–e*). The anterior frontal process of *Sa. harrisi* is longer than those of *Brachyopsemys tingitana* or *Leyvachelys cipadi*, which share the anteroposteriorly short but mediolaterally wide skull proportions of *Sa. harrisi*. However, a relatively long anterior frontal process is present in *Angolachelys mbaxi*, which has a more elongate skull form. Posterolaterally, the frontal contacts the postorbital, and posteriorly the parietal. The lateral margin of the frontal forms the posterodorsal margin of the orbit. Ventrally, the frontal extends underneath the prefrontal, and forms a low and thin crista cranii (figure 4*b*). Due to the low mediolateral width of the anterior frontal process, the crista cranii of both frontals jointly define a narrow, ventrally open sulcus olfactorius, which is similar to that of *Brachyopsemys tingitana* [5].

### 3.1.4. Parietal

The parietal of *Sandownia harrisi* is a large bone that forms the majority of the skull roof and large parts of the braincase via the descending process (figure 3*a*,*c*,*d*,*f*). On the skull roof, the parietal contacts the frontal anteriorly and the extremely elongate postorbital laterally. The posterior margin of the parietal is not entirely preserved in either right or left element, but preserved portions of the original margin on the right side indicate that the temporal emargination of *Sa. harrisi* was slightly larger than that of *Angolachelys mbaxi* [3], *Brachyopsemys tingitana* [5] or *Leyvachelys cipadi* [4]. A minor contact with the squamosal, as present in other sandownids, is preserved on the right side of the skull, as evident in our CT scans and thus confirming the observation of Meylan *et al.* [2] (figure 4*b*). The parietal of *Sa. harrisi* fully overlaps the supraoccipital dorsally with its posterior edge and the supraoccipital is therefore concealed in dorsal view.

The descending process of the parietal of *Sa. harrisi* forms an anteroposteriorly extensive sheet of bone, that contacts the palatine anteroventrally, the pterygoid ventrally and the prootic posteroventrally (figure 3*d*). An unusual feature of *Sa. harrisi* is a small contact of the medial surface of the descending process with the trabecula of the parabasisphenoid at the position of the foramen anterius canalis carotici palatinum, i.e. the anterior exiting foramen for the palatine artery (see below). At its contact with the prootic, the descending process of the parietal forms a posterolaterally and ventrally directed ramus, which extends ventrally underneath the prootic portion of the processus

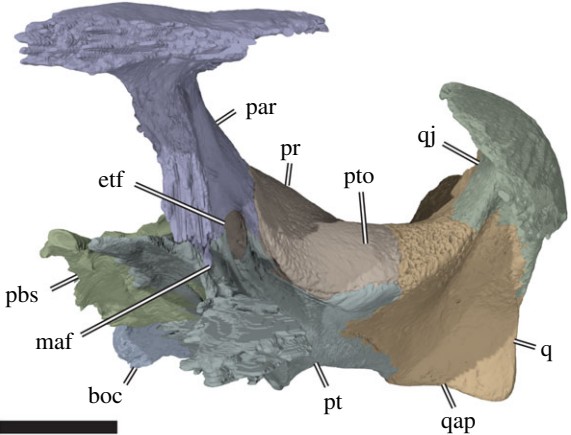

**Figure 5.** Three-dimensional rendering of partial cranium of *Sandownia harrisi* (MIWG 3480) in left anterolateral view, showing aspects of the otic capsule and adjacent structures. boc, basioccipital; etf, external trigeminal foramen; maf, mandibular artery foramen; par, parietal; pbs, parabasisphenoid; pr, prootic; pt, pterygoid; pto, processus trochlearis oticum; q, quadrate; qap, quadrate articular process; qj, quadratojugal. Scale bar, 10 mm.

trochlearis oticum (figure 5). This ramus forms the roof of the cavum epiptericum and excludes the prootic from contributing to the margin of the external trigeminal foramen (*sensu* [18]), which is entirely enclosed by the parietal and the pterygoid. On the dorsal surface of the otic chamber, the parietal forms a small contribution to the deep, ventrally concave processus trochlearis oticum. A similar condition is present in *Solnhofia parsonsi* (TM 4023), in which the respective parietal contribution to the processus trochlearis oticum is, however, much larger.

### 3.1.5. Postorbital

The postorbital of *Sa. harrisi* forms the posterior margin of the orbit, and contacts the frontal anteromedially, the parietal posteromedially, the jugal lateroventrally and the quadratojugal posteroventrally (figure 3*a,c,f*). A contact with the squamosal is apparent from the small squamosal fragment which is preserved on the ventral side of the right temporal roofing (figure 4*b*). The ventral surface of the postorbital has a low, mediolateral ridge, which continues laterally onto the jugal and medially onto the parietal (figure 4*b*). This ridge forms the weak posterior delimitation of the orbit. This ridge is also present, but massively developed, in *Solnhofia parsonsi* (TM 4023). The inferred squamosal/parietal contact implies that the postorbital did not contribute to the temporal emargination.

### 3.1.6. Jugal

The jugal of *Sandownia harrisi* is a large bone at the lateral surface of the skull (figure 3*a–c*). External contacts are present with the maxilla anteriorly, the postorbital dorsally and the quadratojugal posteriorly. A contact with the squamosal is absent. The jugal forms only a small portion of the posteroventral margin of the orbit (figure 3*c*). The ventral margin of the jugal extends to the ventral margin of the maxilla, as in other sandownids but unlike in *Solnhofia parsonsi* (TM 4023; [19]), in which the jugal is positioned relatively more dorsally and is ventrally framed by an extended posterior process of the maxilla. The ventral margin of the jugal of *Sa. harrisi* is concave upward, and forms the anterior half of a moderately developed cheek emargination (figure 3*c*). The cheek emargination of *Sa. harrisi* is deeper than that of other sandownids [3–5].

As in most turtles, the jugal of *Sa. harrisi* has a medial process that braces the lateral skull surface against the palate (figures 3*b* and 4*a*). This process is absent in protostegids [6,8,20,21] and plesiochelyids [6,22,23], both of which are groups with which sandownids have been proposed to be closely related (e.g. [12]). The medial jugal process of *Sa. harrisi* is posteriorly recurved, forming a posteriorly elongate hook-like ramus that extends along the lateral surface of the pterygoid (figures 3*b* and 4*a*). Both the recurved shape and posterior extent are similar to the jugal morphology of *So. parsonsi* (TM 4023; [19]) and probably also *Angolachelys mbaxi* [3], for which the jugal has not been described in detail. As in *So. parsonsi*, the medial jugal process of *Sa. harrisi* additionally contacts the

palatine dorsally, as well as the maxilla ventrally. The dorsal surface of the medial jugal process forms a concave floor to the orbit in both taxa, but does not contribute to the foramen orbito-nasale due to a small contact of the palatine with the maxilla along its margin (figure 4a). *Sandownia harrisi* and *So. parsonsi* furthermore share a thin, almost lamina-like ridge on the posterodorsal and medial surface of the medial jugal process, which delimits the orbital cavity from the adductor chamber, and which continues dorsally onto the internal surface of the postorbital. The posterior surface of the medial jugal process is deeply recessed between the laminar ridge just described and the lateral surface of the skull in *Sa. harrisi* and *So. parsonsi*. In both taxa, there is a weak and superficial groove on the lateral surface of medial jugal process, which leads anteriorly to a small foramen, the foramen supramaxillare (figure 4a). This placement is unusual, as this foramen is usually formed by the maxilla. The canalis infraorbitalis extends through the jugal and enters the maxilla, where it supplies the palate. In *Sa. harrisi*, the medial process of the jugal contributes to the ventrally exposed roof of the secondary palate (figure 3b). It is possible that this part of the palate is not part of the functional triturating surface, because large neurovascular foramina that indicate the approximate position of a keratinous rhamphotheca are limited to the maxilla, premaxilla and vomer (see below). In *So. parsonsi*, the medial process of the jugal does not extend this far ventrally, so that the jugal is excluded from the ventral palatal surfaces. However, a jugal contribution to this surface is evident in *B. tingitana* [5]. In *B. tingitana* the posteriorly recurved ramus of the medial jugal process does not extend as far along the pterygoid as in *Sa. harrisi* or *So. parsonsi*. Additionally, the ventral surface of the medial process of the jugal forms peculiar socket-like depressions on the palate in *B. tingitana*, but these structures are clearly absent in *Sa. harrisi* or *So. parsonsi*.

### 3.1.7. Quadratojugal

The quadratojugal of *Sa. harrisi* forms the posterior portion of the lateral surface of the skull (figure 3a–c). On the skull surface it contacts the jugal anteriorly, the postorbital anterodorsally, the squamosal dorsally and the quadrate posteroventrally and medially. The ventral margin of the quadratojugal is concavely curved, forming the posterior portion of a moderately deep cheek emargination. Posteroventrally, the quadratojugal extends as a thin spur along the quadrate. The quadratojugal does not contribute to the anterior margin of the cavum tympani or the margin of the temporal emargination. On the medial surface, the quadratojugal forms a short but stout process that articulates with the quadrate, and forms the lateralmost part of the mediolaterally broad, and deeply concave processus trochlearis oticum (figures 4a and 5). A clear quadratojugal contribution to the trochlea process is unusual for turtles, but present in *Solnhofia parsonsi* (TM 4023), *Angolachelys mbaxi* (based on reconstructions in [3]) and *Brachyopsemys tingitana* (based on CT scans of AMNH 30612). It is unclear if this feature is also present in other sandownid *Leyvachelys cipadi*. In all other turtles, this process with such a clear contribution to the processus trochlearis oticum is absent (contra [24], who identified a minor contribution in some trionychids). The contact of the quadratojugal with the postorbital is not preserved in *Sa. harrisi*, but can be inferred based on the preserved sutures.

### 3.1.8. Squamosal

The squamosal is not fully preserved on either side of the skull in MIWG 3480 (see also [2]). However, two fragments of the right squamosal are preserved: a small fragment is preserved ventral to the posterior end of the parietal and postorbital, showing that contacts with both these bones exist at this place (figure 4b). A broad, quadratojugal-squamosal contact is additionally seen on the internal side of the right side of the skull just above the middle ear (figure 4a). The quadrates show clearly that the squamosal would have formed parts of the antrum postoticum.

### 3.1.9. Premaxilla

Meylan *et al.* [2] interpret the premaxillae of *Sandownia harrisi* to be anteriorly fused, but we could trace an interpremaxillary suture through the entire contact zone of the left and right premaxillae. The premaxilla is a relatively small bone at the anterior end of the snout, which contacts the maxilla along its posterolateral surface, and the vomer posteriorly (figure 3a–c,e). Each premaxilla has a concavely incised anterodorsal margin and jointly form a dorsally raised median subdivision of the external nares. Together, the right and left premaxillae form a low dorsal mount in the ventral margin of the external naris, giving this opening an inverted heart-shaped outline. The surface of the premaxilla

along the external contact with the maxilla is gently recessed to form a shallow vertical trough between premaxilla and maxilla. The anteroventral margin of the premaxilla forms a low and blunt labial ridge that is continuous with the respective ridge on the maxilla (figure 3*b*). The anterior and ventral surfaces of the premaxilla are perforated by numerous small neurovascular foramina, indicating the presence of a keratinous rhamphotheca in *Sa. harrisi*. There is no evidence for the presence of prepalatine foramina.

### 3.1.10. Maxilla

The maxilla of *Sandownia harrisi* is a large bone, forming most of the upper jaw (figure 3*b,c,e*). It contacts the premaxilla anteroventrally, the vomer ventromedially, the nasal dorsomedially, the prefrontal dorsally, the palatine posteromedially, the pterygoid posteroventrally and the jugal posterolaterally. The maxillary contribution to the orbit is much smaller in *Sa. harrisi* than in *Solnhofia parsonsi*, but not absent as reported for *Brachyopsemys tingitana* [5]. The maxilla of *Sa. harrisi* also forms only a minor part of the ventral surface of the orbital cavity; a thin bar of the maxilla is wedged between the prefrontal anteriorly and the jugal posteriorly (figure 3*a*). However, this bar of bone forms the medial margin of the large foramen orbito-nasale and extends posterior to contact the palatine, which forms the posterior margin of this foramen. The lateral surface of the maxilla is dorsoventrally nearly as deep as it is anteroposteriorly long. In other sandownids, the maxilla has been reported to be dorsoventrally taller than anteroposteriorly long [3–5], whereas it is usually longer than tall in most turtles, including *So. parsonsi* (TM 4023). A moderately deep, ventrally weakly convex sulcus extends mediolaterally over the external maxillary surface (figure 3*c,e*). As stated by Meylan *et al.* [2], this sulcus probably delimits the dorsal margin of the keratinous rhamphotheca on the maxilla. The low and rounded labial ridge of the maxilla, as well as the ventral palatal surface of the triturating surface, is covered in neurovascular foramina (figure 3*b*). The neurovascular foramina are larger than those of the premaxilla and vomer. An intermaxillary contact, as reported for *Brachyopsemys tingitana* [5] is absent in *Sa. harrisi* due to the presence of a ventrally exposed vomer that contacts the premaxillae (see below).

### 3.1.11. Vomer

The vomer of *Sandownia harrisi* is an unpaired median element of the palate (figure 3*b,d*). The ventral process is relatively large in *Sa. harrisi* and ventrally well exposed, similar to that of *Angolachelys mbaxi* [3]. The vomer of *Brachyopsemys tingitana* [5], *Leyvachelys cipadi* [4] and also *Solnhofia parsonsi* [6,12,19] are different in that the exposure of their ventral process on the palate is extremely reduced by the maxillae, which overlap the vomer ventrally and nearly contact one another medially (figure 3*b*). However, the vomer of all taxa mentioned above share an anteroposteriorly oriented, longitudinal furrow that extends along the entire ventral surface of the vomer. In taxa with mediolaterally narrow vomers, such as *So. parsonsi*, the furrow becomes extremely constricted. Data from *L. cipadi*, for which multiple skulls of different sizes exist [17], indicate that the exposure of the vomer on the palate becomes reduced in larger skulls, suggesting that the differences described above among various taxa may represent ontogenetic, not taxonomic differences. This is supported by the fact that the skull of *B. tingitana*, which shows the smallest ventral exposure of the vomer among the mentioned taxa, represents the overall largest individuals. The ventral process of the vomer contacts the premaxilla anteriorly, the maxilla anterolaterally, the palatine posterolaterally, and forms the ventral margin of the internal nares.

Anterodorsally, the vomer of *Sa. harrisi* has a columnar dorsal process, which expands dorsolaterally to contact the left and right prefrontal anteriorly and the palatine posteriorly. Anterior to the prefrontal contact, the vomer has a surface that slopes anteroventrally and floors parts of the nasal cavity (figure 3*d*). A low median ridge extends onto this surface anteriorly from the fissure ethmoidalis, and probably served as the insertion for an internarial septum during life [18]. From the posterior surface of the dorsal process of the vomer, a lamina extends posteriorly toward the internal naris. This lamina forms the median separation between the left and right choanal tunnels, which connect the nasal cavity with the internal nares. The vomer does not contribute to the medial margin of the foramen orbito-nasale, as the maxilla and palatine contact one another along the medial surface of the vomer.

### 3.1.12. Palatine

 The palatine of *Sandownia harrisi* is a paired element of the posterior palate, and forms large parts of the narial passages and internal choanae. On the ventral surface of the skull, the palatine contacts the maxilla

anteriorly, the vomer medially and the pterygoid laterally and posteriorly (figure 3b). The palatine contributes to the ventral and lateral margins of the internal naris, which is developed as a wide, singular opening with a posteriorly concave margin, as in other sandownids. The morphology of the posterior part of the secondary palate is generally similar to that of *Solnhofia parsonsi*, although the palatines in the latter are mediolaterally narrower, as is the exposure of the vomer. In *So. parsonsi*, a posterior process of the palatine extends from the margin of the internal naris along the medial surface of the pterygoid, forming the lateral wall of the posterior extension of the choanae. The foramen posterius palatinum of *So. parsonsi* is completely formed by the palatine at the basis of this process. According to Meylan *et al.* [2], a similar, albeit much thinner process is found in the equivalent position in *Sandownia harrisi*, forming the medial margin of the foramen palatinum posterius. However, although this spur-like process seems to have a superficial suture with the pterygoid, the CT scans indicate that this spur is formed by the pterygoid. It is difficult to see in the CT scans if the respective process is truly part of the pterygoid, or part of the adjacent palatine, and we therefore do not want to dismiss the interpretation by Meylan *et al.* [2] at this point. The foramen palatinum posterius is completely located within the palatine in *Leyvachelys cipadi* [4], but is formed between the palatine and pterygoid in *Brachyopsemys tingitana* [5]. It has not been indicated to be present in the description of *Angolachelys mbaxi* [3].

The dorsal plate of the palatine of *Sandownia harrisi* roofs the narial passage (figures 3d and 4a). It forms the medial margin of the foramen orbito-nasale, where it has a small contact with the maxilla, and an even smaller contact with the prefrontal anteriorly. Posterior to the dorsal contact with the vomer, the right and left palatines form a median contact (figure 4a). The palatines posterodorsally contact the pterygoids on the floor or the cavum cranii. Just anterior to the palatine-pterygoid contact, there is a foramen that extends from the cavum cranii into the narial passage, just anterior to its opening into the internal naris (figure 4a). The foramen has, to our knowledge, not been observed or described for any turtle before. It is a relatively large foramen, comparable to the size of the opening of foramina associated with the carotid articulation. The foramen anterius canalis carotici palatinum (faccp), i.e. the anterior exiting foramen for the palatine artery, is positioned a short distance posterior to the position of the unnamed foramen. Therefore, it seems possible that the palatine artery, after entering the cavum cranii, extends anteriorly to the foramen and then sends off a branch that enters the choanal tunnel to supply blood to this region of the skull. In *So. parsonsi* (TM 4023), this region of the palatine-pterygoid contact is broken, so that it is not possible to see if a similar foramen is present in this taxon.

The dorsal part of the palatine of *Sandownia harrisi* also has a short ascending process, which forms the anterior margin of the lateral wall of the braincase, and contacts the parietal via an oblique, posteroventrally sloping suture (figure 3d). A dorsal process of the palatine is absent in *So. parsonsi*.

### 3.1.13. Quadrate

The quadrate of *Sandownia harrisi* forms the cavum tympani, most of the antrum postoticum, and the enclosed incisura columellae auris, contributes to the processus trochlearis oticum, forms the lateral wall of the cavum acustico-jugulare and the canalis stapedio-temporale, and forms the articular process for the mandible. The quadrate contacts are as stated in Meylan *et al.* [2], with the opisthotic posteriorly and posteromedially, the prootic anteromedially, the pterygoid ventromedially, the quadratojugal anterolaterally and the squamosal posterodorsally.

The short, but deep cavum tympani is anteriorly formed entirely by the quadrate (figure 3c), without a contribution from the quadratojugal. The cavum tympani was posterodorsally bound by the squamosal, which is not preserved in this region of the skull. The antrum postoticum must have been relatively extensive, based on the large opening along the posterodorsal margin of the quadrate (figure 3c), onto which the squamosal would have been positioned if it were preserved more completely. The incisura columellae auris is completely enclosed by the quadrate, unlike in *Solnhofia parsonsi* (TM 4023; [19]) or *Plesiochelys planiceps* (OUMNH J1582), in which the incisura remains open posteroventrally. As noted by Meylan *et al.* [2], the closure of the incisura in *Sandownia harrisi* seems to be the result of the quadrate being sutured to itself along the posteroventral aspect of the incisura. Dorsal to the suture, there is a very narrow groove that extends from the cavum tympani to the cavum acustico-jugulare, which likely to represent the groove for the Eustachian tube, as also observed in some bothremydid pleurodires [25,26]. The infolding ridge of the quadrate, a prominent lip of bone on the posterior surface of the mandibular process [9] that is present in thalassochelydians (*sensu* [26], including plesiochelyids and *Solnhofia parsonsi*) but also sandownids [6], is much less

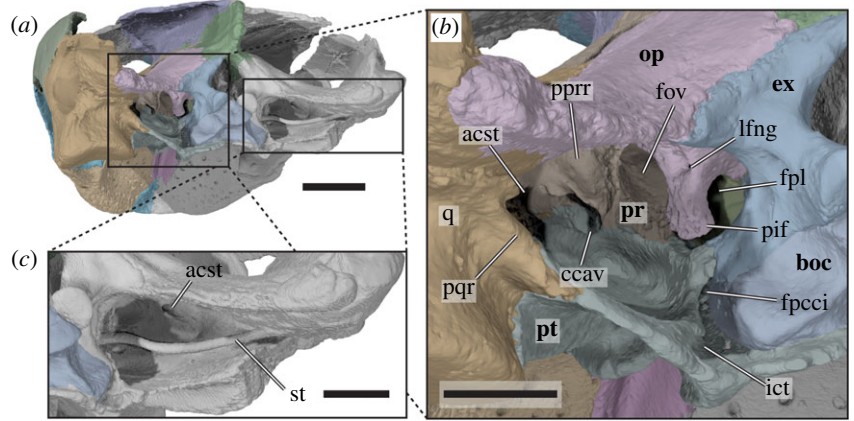

**Figure 6.** Three-dimensional renderings of the posterior part of the cranium forming the cavum acustico-jugulare of *Sandownia harrisi* (MIWG 3480). (*a*) Cranium in posterior and slightly left lateral view. (*b*) Close-up of left posterior side of cranium, showing the cavum acustico-jugulare and associated structures. (*c*) Close-up of right cavum acustico-jugulare, showing articulated stapes. acst, aditus canalis stapedio-temporale; boc, basioccipital; ccav, canalis cavernosus; ex, exoccipital; fov, fenestra ovalis; fpcci, foramen posterius canalis carotici internus; ict, internal carotid trough; pif, processus interfenestralis; fpl, fenestra perilymphatica; lfng, lateral foramen nervi glossopharyngei; op, opisthotic; pprr, posterior prootic recess; pqr, posterior quadrate ridge; pr, prootic; q, quadrate; st, stapes. Scale bar in *a* equals 10 mm, scale bars in *b* and *c* equal 5 mm.

prominent in *Sandownia harrisi* than in *Brachyopsemys tingitana* (AMNH 30612). In *Sandownia harrisi* it is limited to the medial part of the posterior surface of the mandibular process of the pterygoid (figure 6), where it connects to the posterior margin of the pterygoid, as is also the case in *Brachyopsemys tingitana* (AMNH 30612), *Plesiochelys planiceps* (OUMNH J1582) and *Solnhofia parsonsi* (TM 4023).

The facet for the mandibular articulation is anteroposteriorly broader laterally than medially in *Sandownia harrisi* and anteroposteriorly incised by a shallow sulcus that defines medial and lateral subfacets. The mandibular process of the quadrate is relatively low, and the pterygoid contacts the quadrate very close to the medial margin of the mandibular facet.

The anterior surface of the quadrate slopes dorsally from the mandibular facet to the processus trochlearis oticum, of which the quadrate only forms the lateral third, as in *Solnhofia parsonsi* (TM 4023), but unlike in *Plesiochelys planiceps* (OUMNH J1582), in which the quadrate contribution is much more extensive and forms more than half of its width. Unlike in *Solnhofia parsonsi* and *Plesiochelys planiceps*, the facet of the otic trochlear process spans over the entire width of the otic process in *Sandownia harrisi*. However, *Solnhofia parsonsi* and *Sandownia harrisi* share a short but distinct quadratojugal contribution to the processus trochlearis oticum. Long epipterygoid processes are absent in both these taxa, whereas such a process is seen in *Plesiochelys planiceps* (OUMNH J1582).

The dorsal surface of the quadrate forms large portions of the otic capsule in *Sandownia harrisi* and *Solnhofia parsonsi*. The quadrate only has a small superficial contribution to the foramen stapedio-temporale in *Sandownia harrisi* (figure 4*a*), but it forms the lateral wall of the stapedio-temporal canal as in most turtles.

### 3.1.14. Epipterygoid

Unlike *Solnhofia parsonsi* (TM 4023) or plesiochelyids (e.g. *Plesiochelys planiceps*: OUMNH J1582), *Sandownia harrisi* has no epipterygoid (figure 5). For *Leyvachelys cipadi*, epipterygoids have been reported [17].

### 3.1.15. Pterygoid

The pterygoid of *Sandownia harrisi* is a large bone at the interface between palate and basicranium (figure 3*b*,*d*), thereby contributing to many structures, as in all turtles. However, it is also peculiar in several features. As in most non-pleurodiran turtles, the pterygoid is anteroposteriorly elongate and contributes to the ventral flooring of the cavum acustico-jugulare, the formation of the foramen for the maxillomandibular division of the trigeminal nerve (i.e. the external trigeminal foramen) and the secondary lateral braincase in which this foramen is positioned, houses parts of the carotid artery

canals, and forms parts of the secondary palate. Some of the more unusual features of the pterygoid of *Sandownia harrisi* are the presence of an additional (mandibular artery?) foramen anteroventrally to the position of the external trigeminal foramen, a well-developed ventral ridge that is similar to the lateral flanges of the pterygoid seen in podocnemidid pleurodires (e.g. Gaffney *et al*. [27] for many illustrations of podocnemidids), a fenestra caroticus, an articulation socket for the basipterygoid process, a contribution to the processus trochlearis oticum, and a deep, pocket-like fossa within the floor of the cavum acustico-jugulare. These structures are described in more detail below.

As in most turtles, the pterygoid of *Sandownia harrisi* contacts the quadrate posterolaterally, the prootic dorsomedially behind the external trigeminal foramen, the parietal dorsally along the crista pterygoidei, the jugal along the medial margin of the subtemporal fenestra, the palatine anteriorly and medially, and the parabasisphenoid and basioccipital medially. An additional contact is present with the exoccipital; this is variously developed among turtles in general, and is for instance present in *Plesiochelys planiceps* (OUMNH J1582) and *Plesiochelys etalloni* (MNB 435), but absent in *Solnhofia parsonsi* (TM 4023). Another contact of the pterygoid that is not universally present in turtles but developed in *Sandownia harrisi* is with the maxilla along the anterior margin of a very elongate but mediolaterally narrow process that extends within the palate and contributes to the triturating surface (figure 3*b*). A maxilla-pterygoid contact is also present in *Solnhofia parsonsi*, but not in protostegids [8] or plesiochelyids (e.g. *Plesiochelys etalloni*: [28]; *Plesiochelys bigleri*: [23]; *Plesiochelys planiceps*: [6]), with the exception of *Portlandemys mcdowelli* [28].

The anterior process of the pterygoid of *Sandownia harrisi* is tightly interfingered with the posterior process of the jugal, as in *Solnhofia parsonsi* (TM 4023), but different from plesiochelyids, in which the jugal lacks a medial process [6]. Together with the jugal, the anterior process of the pterygoid forms the medial margin of the subtemporal fenestra in *Sandownia harrisi* (figures 3*b* and 4*a*). In most non-pleurodiran turtles, the lateral margin of the pterygoid forms a lateral process in this region, the processus pterygoideus externus. This process is particularly clearly developed in plesiochelyids (e.g. [6,28,29]). *Sandownia harrisi*, on the other hand, bears no processus pterygoideus externus [2], similar to the condition in other sandownids or some trionychians [24], but also *Solnhofia parsonsi*, in which the process is extremely reduced [19]. The anterior process in *Sandownia harrisi* extends anteriorly along more than one third of the length of the triturating surface until it contacts the maxilla. However, the pterygoid contribution to this palatal surface is positioned posterior to the level at which the large neurovascular foramina seen in the maxillae and vomer disappear (figure 3*b*). This suggests that this part of the secondary palate was perhaps not covered by the rhamphotheca, and was thus not part of the functional part of the triturating surface that processed food items. Along its medial margin, the anterior process of the pterygoid contacts the palatine. As noted in the description of the palatine (see above), the pterygoid seems to form a short, anteromedially directed spur that forms the medial margin of the small foramen palatinum posterius, although it cannot be ruled out entirely that this spur of bone was formed by the palatine (figure 3*b*).

The anterodorsal process of the pterygoid of *Sandownia harrisi* is dorsally arched above the level of the internal naris (figure 3*d*) and forms the roof of the nasal passage. It contacts the palatine anteriorly and the other pterygoid medially. This extended interpterygoid contact floors the anterior part of the parabasisphenoid, as in most turtles. However, the parabasisphenoid has only a small median exposure behind this suture, as the pterygoids form a second interpterygoid contact with their posterior processes. This posterior interpterygoid contact also strongly reduces the ventral exposure of the basioccipital (figure 3*b*), a feature that is both unique to and synapomorphic for sandownids [4,6]. Just anterolateral to the parabasisphenoid exposure, the ventral surface of each pterygoid bears a deep fossa (figures 3 and 7*a*), which is identified as the fenestra caroticus (as already stated by [30]). A fenestra caroticus is absent in *Solnhofia parsonsi* (TM 4023), whereas plesiochelyids show a great deal of variation in this regard: In *Plesiochelys etalloni* (MNB 435), a fenestra postoticus is clearly absent, and the entire carotid arterial system is encased within the basicranium. However, in other specimens of the same species (e.g. NHMS 400870, NMS 40871), a fenestra caroticus is clearly present, though not labelled as such [9,28].

Posterior to the fenestra caroticus, the posterior process of the pterygoid of *Sandownia harrisi* forms two distinct ridges. One of these ridges develops from the lateral margin of the anterior pterygoid process and extends posteriorly to the margin of the fenestra postotica at the back of the skull. The same ridge is apparent in *Solnhofia parsonsi* (TM 4023), in which the ridge is more shallowly developed. The pterygoid ridge of *Sandownia harrisi* is particularly prominent approximately at mid-length, where it extends ventrally to form a flange (figure 3*b*). A similar flange is developed in other turtles, such as carettochelyids (e.g. [31]: NHMUK 1903.7.10.11). At the level of this flange, a second ventral ridge diverges posterolaterally and extends toward the mandibular process of the quadrate.

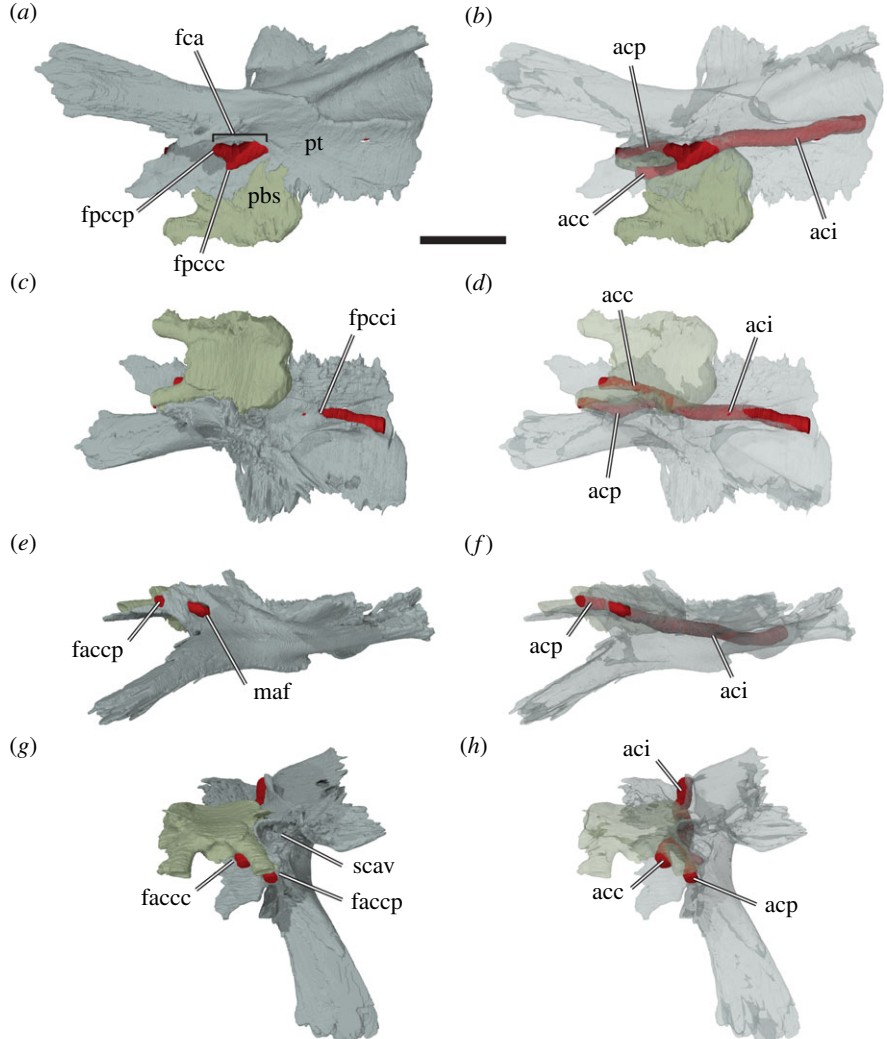

**Figure 7.** Three-dimensional renderings of the left pterygoid, parabasisphenoid, and interpreted carotid artery of *Sandownia harrisi* (MIWG 3480) in different views. (*a*) Bones rendered solid in ventral view. (*b*) Bones rendered transparent in ventral view. (*c*) Bones rendered solid in posterodorsal view. (*d*) Bones rendered transparent in posterodorsal view. (*e*) Bones rendered solid in left lateral view. (*f*) Bones rendered transparent in left lateral view. (*g*) Bones rendered solid in anterodorsal view. (*h*) Bones rendered transparent in anterodorsal view. acc, cerebral artery; aci, internal carotid artery; acp, palatine artery; fca, fenestra caroticus; faccc, foramen anterius canalis carotici cerebralis; faccp, foramen anterius canalis carotici palatinum; fpccc, foramen posterius canalis carotici cerebralis; fpcci, foramen posterius canalis carotici internus; fpccp, foramen posterius canalis carotici palatinum; maf, foramen for the mandibular artery; pbs, parabasisphenopid; pt, pterygoid; scav; sulcus cavernosus. Note that the carotid artery has been reconstructed by segmenting the interior of the carotid canals. Scale bar, 10 mm.

Both pterygoid ridges form the margin of the deeply developed pterygoid fossa, which extends over nearly the full length of the posterior pterygoid process.

Through the fenestra caroticus of *Sandownia harrisi*, the basipterygoid process of the parabasisphenoid is exposed [30]. This process, although distinct, is tightly fit into a pocket within the medial surface of the pterygoid (figure 8). A basipterygoid process is clearly absent in *Solnhofia parsonsi* (TM 4023) and *Plesiochelys planiceps* (OUMNH J1582), as is evident from CT scans and 3D models [16]. Within the fenestra caroticus of *Sandownia harrisi*, the canals for the cerebral artery and the palatine artery extend anteromedially and anterolaterally, respectively, and the canal for the internal carotid artery extends posteriorly. The foramina posterius canalis caroticus palatinum (fpccp) and posterius canalis carotici cerebralis (fpccc) lie mediolaterally adjacent to one another (figure 7), as in *Plesiochelys etalloni* [29]. The palatine artery canal of *Sandownia harrisi* is formed within the parabasisphenoid-pterygoid suture, as in most turtles, although the pterygoid has a larger contribution to the canal wall. This canal seems to be fully confluent with the canalis nervus vidianus (*sensu* [32]), as no separate branch is apparent

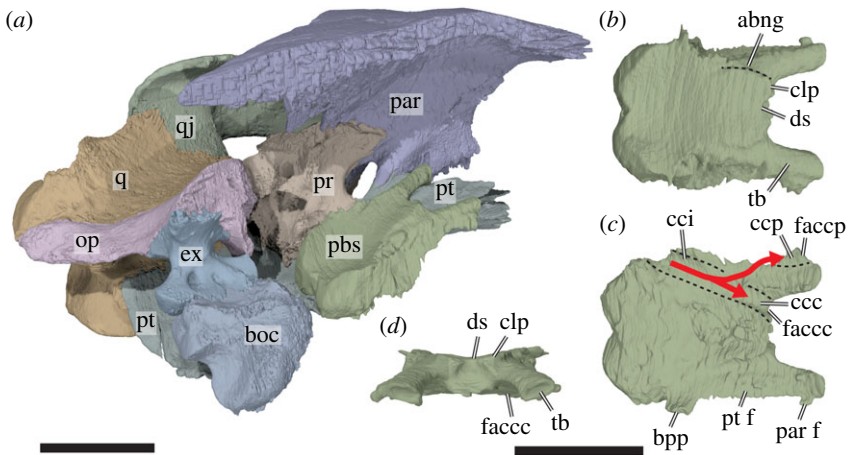

**Figure 8.** Three-dimensional renderings of the parabasisphenoid and surrounding bones of *Sandownia harrisi* (MIWG 3480). (*a*) Posterodorsomedial view of partial left basicranium. (*b*) Parabasisphenoid in dorsal view. (*c*) Parabasisphenoid in ventral view. (*d*) Parabasisphenoid in anterior view. abng, abducens nerve groove; bpp, basipterygoid process; cci, canalis carotici internus; ccc, canalis carotici cerebralis; ccp, canalis carotici palatinum; clp, clinoid process; ds, dorsum sellae; ex, exoccipital; faccc, foramen anterius canalis carotici cerebralis; faccp, foramen anterius canalis carotici palatinum; op, opisthotic; par, parietal; par-f, parietal facet; pbs, parabasisphenoid; pr, prootic; pt, pterygoid; pt-f, pterygoid facet; q, quadrate; qj, quadratojugal; tb, trabecula. Dashed lines in *b* indicate the course of the groove for the abducens nerve. Dashed lines in *c* indicate the course of carotid artery branches. Arrow in *c* illustrates course of the internal carotid artery and its branches. Scale bar, 10 mm.

for the anterior division of the facial nerve. The cerebral artery canal, however, is largely formed by the pterygoid, and the parabasisphenoid only roofs the canal (figure 7). This is highly unusual, as the cerebral artery usually penetrates the parabasisphenoid only (i.e. without pterygoid contributions) in all amniotes including turtles (e.g. [33]). The cerebral canal exits into the braincase immediately medial to and at the base of the parabasisphenoid trabeculae, somewhat similar to *Dermochelys coriacea* ([16]: UMZC R3031), in which the cerebral artery also extends along the ventrolateral margin of the parabasisphenoid without being fully encased by it. The exiting foramen for the palatine artery, the foramen anterius canalis carotici palatinum (faccp), is in a more commonly found position (e.g. see protostegids: [8]; or *Solnhofia parsonsi*) ventral to the sulcus cavernosus (figure 7*g*,*h*). However, the foramen anterius canalis caroticus palatinum of *Sandownia harrisi* is unusual in that besides the pterygoid, which forms the ventral part of the margin of the foramen, the parabasisphenoid and also the parietal contribute to the foramen (figure 8). The parietal has only a short lateral contribution, while the trabeculae of the parabasisphenoid form the mediodorsal margin of the faccp (figure 7*g*).

The course of the internal carotid artery posterior to its split can be traced through the posterior process of the pterygoid in CT scans. The canalis caroticus internus extends posteriorly up to the level of the recessus scalae tympani, which is predominantly formed by the exoccipital and opisthotic, but which is ventrally floored by the pterygoid. The internal carotid canal opens here to form the foramen posterius canalis carotici interni (fpcci) on the dorsal surface of the pterygoid (figures 6*a*,*b* and 7*c*). In *Solnhofia parsonsi* (TM 4023) the fpcci opens within the posterior margin of the pterygoid and thus is a more ventral position than in *Sandownia harrisi*. Regarding the position of the fpcci, *Solnhofia parsonsi* and *Plesiochelys etalloni* are more similar to one another than either is to *Sandownia harrisi*. The course of the internal carotid artery can still be traced in *Sandownia harrisi* posterior to the fpcci, because the dorsal surface of the pterygoid is deeply incised by a dorsally open trough that has the same width as the internal carotid canal diameter (figures 6*a*,*b* and 7*c*). Medial to this carotid trough, the basioccipital articulates with the dorsal surface of the posterior process of the pterygoid. On the lateral side of the carotid canal trough, there is a vertically ascending lamina of the pterygoid, which extends from the posterior margin of the pterygoid anteriorly to the prootic (figure 6*b*). This lamina forms the medial margin of the floor of the cavum acustico-jugulare. Thus, the carotid trough of *Sandownia harrisi* is different to a similar trough for the carotid artery seen in testudinoids, kinosternids and some chelydrids as well as some chelonioids (see character 149 of [6]), which is positioned more medially within the cavum acustico-jugulare and is dorsally roofed by the prootic within the cavum. The dorsal pterygoid lamina of *Sandownia harrisi* has its dorsally highest extent

posteriorly at the level of the exoccipital, with which it contacts along a well-developed, interdigitated suture. Anterior to the level of the exoccipital, the lamina continues ventral to the level of the processus interfenestralis of the opisthotic and the fenestra ovalis, which is formed by this process of the opisthotic posteriorly, and a ventral process of the prootic anteriorly. The pterygoid lamina connects to this ventral process of the pterygoid, but because it does not contact the opisthotic, the pterygoid does not truly form the ventral margin of the fenestra ovalis, as originally stated by Meylan *et al*. [2]. However, the pterygoid does form the floor of the inner ear cavity. The margin of the pterygoid lamina is recurved laterally in its anterior part, and turns posterolaterally at the level of the prootic contact, thereby circumscribing a large, pocket-like fossa within the floor of the cavum acustico-jugulare (figure 6). This extension of the cavum acustico-jugulare is a very unusual feature of *Sandownia harrisi* and not seen in *Solnhofia parsonsi* or plesiochelyids. However, an extremely wide fenestra postotica is also present in *Leyvachelys cipadi* [17] and, seemingly, in *Angolachelys mbaxi* [3]. Anterolateral to this fossa within the cavum acustico-jugulare, and slightly dorsally elevated from its floor, the pterygoid forms the floor of the posterior foramen for the canalis cavernosus in *Sandownia harrisi*. The canalis cavernosus remains floored by the pterygoid for its entire length and is roofed by the prootic. The lateral wall of the canalis cavernosus is formed by the prootic process of the pterygoid. Halfway along the canalis cavernosus, and directly ventrally to the position of the medial foramen for the facial nerve (CN VII) that is enclosed within the prootic, there is a short, ventromedially directed canal. This canal is the foramen pro ramo nervi vidiani, and connects the canalis cavernosus with the canal for the internal carotid artery [34]. The foramen pro ramo nervi vidiani carries the vidian (= palatine) branch of the facial (CN VII) nerve [35,36]. At the anterior end of the prootic process of the pterygoid, the external trigeminal foramen opens on the lateral side of the canalis cavernosus (figure 5). Anterior to the foramen cavernosus, the dorsally open continuation of the canalis cavernosus is referred to as the sulcus cavernosus [34]. The sulcus cavernosus extends only for a short distance anteriorly to the level of the external trigeminal foramen, and is formed in the contact between the pterygoid and the trabeculae of the parabasisphenoid.

At the anterior end of the sulcus cavernosus, the crista pterygoidei of *Sandownia harrisi* ascends to meet the parietal. The crista pterygoidei is a relatively low process in *Sandownia harrisi* (figure 5), and lower than the palatine contribution to the secondary lateral wall of the braincase, which it contacts at its anterior margin. In *Plesiochelys planiceps* (OUMNH J1582) and *Solnhofia parsonsi* (TM 4023), the crista pterygoidei is relatively higher, and is laterally overlain by the epipterygoid, which is absent in *Sandownia harrisi*. The crista pterygoidei of *Sandownia harrisi* dorsally contacts the descending process of the parietal and forms the anteroventral margin of the external trigeminal foramen (figure 5). The posteroventral margin of the external trigeminal foramen is formed by the prootic process of the pterygoid. A conspicuous feature of the crista pterygoidei in *Sandownia harrisi* is a large mediolaterally directed opening or short canal, which connects the subtemporal fossa laterally with the canal for the palatine artery (figure 5). It seems likely that the mandibular artery exits through this foramen: the foramen is exceptionally large for a nerve canal, and its direct connection to a structure related to the carotid arterial system further suggests a blood supply function. Although the arterial patterns of extant turtles have been investigated [35,36], and several generalized patterns have been synthesized from these data (e.g. [37]), it is clear that there is substantial variation to the pattern of arterial blood supply among turtles. This makes it difficult to assess the arterial system for turtles that belong to entirely extinct groups, such as *Sandownia harrisi*, especially when previously undescribed foramina like the one described here are found. Whereas the mandibular artery exits the skull via the foramen stapedio-temporale in pleurodires, and through the external trigeminal foramen in most cryptodires [35–37], it has been reported that the mandibular artery occupied the palatine canal in trionychians, in which the palatine artery is absent [35–37]. Based on the identification of the foramen anterius canalis carotici palatinum, it can be inferred that the palatine artery is not reduced in *Sandownia harrisi*. However, possibly the mandibular artery did extend through the same canal as the palatine artery, and then exited through its own foramen. It is noteworthy that this potential mandibular artery foramen of *Sandownia harrisi* is not identical to the foramen identified by Evers & Benson [6] (ch. 124), because that foramen, although possibly also for the mandibular artery, opens from the sulcus cavernosus and not from the palatine artery canal.

The prootic process of the pterygoid forms a ventral bracing to the overlying prootic (figure 5). It is not unusual in turtles that the pterygoid extends dorsally toward the processus trochlearis oticum. However, in *Sandownia harrisi*, the pterygoid extends all the way into the deeply mediolaterally concave trochlea, and forms parts of the lateral margin of this structure.

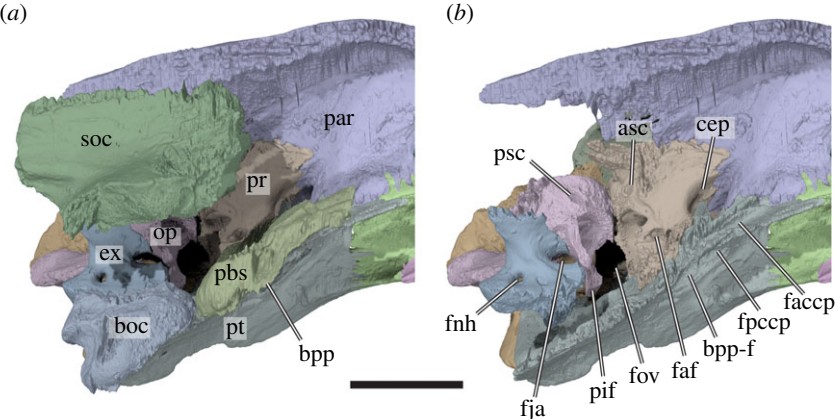

**Figure 9.** Three-dimensional renderings of a medial view of the left side of the basicranium of *Sandownia harrisi* (MIWG 3480). (*a*) Rendering including the supraoccipital. (*b*) Rendering excluding the supraoccipital, parabasisphenoid and basioccipital to show basicranial details. asc, anterior semicircular canal; boc, basioccipital; cep, cavum epiptericum; bpp, basipterygoid process; bpp-f, basipterygoid process facet; ex, exoccipital; faccp, foramen anterius canalis carotici palatinum; faf, fossa acustico-facialis; fja, foramen jugulare anterius; fnh, foramen nervi hypoglossi; fov, fenestra ovalis; fpccp, foramen posterius canalis carotici palatinum; op, opisthotic; par, parietal; pbs, parabasisphenoid; pif, processus interfenestralis; pr, prootic; psc, posterior semicircular canal; pt, pterygoid; soc, supraoccipital. Scale bar, 10 mm.

## 3.1.16. Supraoccipital

The supraoccipital of *Sandownia harrisi* forms the posterior end of the braincase (figures 3*a,d,f*, 4*b* and 9). It is dorsally completely covered by the parietals and projects ventrolateral processes to either side of the skull midline that contact the prootic anteriorly and the opisthotic posteriorly. On the posterior end of the ventral process, the supraoccipital contacts the exoccipital and roofs the foramen magnum.

The ventral process of the supraoccipital houses the common crus and portions of the anterior and posterior semicircular canals of the endosseous labyrinth. The ventral surface between the ventral processes is mediolaterally convex, roofing the braincase. The supraoccipital contributes to the foramen magnum.

The supraoccipital crest of *Sandownia harrisi* shows damage, but likely did not extend posteriorly far beyond the level of the posterior end of the squamosals and is shorter than that of *Plesiochelys planiceps*.

## 3.1.17. Exoccipital

The exoccipital of *Sandownia harrisi* forms the lateral margin of the foramen magnum (figures 3*f* and 6). Dorsally, it forms a broad, dorsolaterally directed process that contacts the supraoccipital and opisthotic. The central part of the exoccipital is columnar and forms the posterior wall of the recessus scalae tympani. Near its base, the columnar part of the exoccipital is pierced by two roughly mediolaterally directed canals for the hypoglossal (XII) nerve. The ventral part of the exoccipital is expanded to form a broad and elongate foot, which contacts the basioccipital. This ventral process extends ventrolaterally to contact the pterygoid (figure 6*b*), which is not the case in *Solnhofia parsonsi* (TM 4023) or *Plesiochelys planiceps* (OUMNJ J1582). The exoccipital footplate of *Sandownia harrisi* and *Solnhofia parsonsi* (TM 4023) is dorsally recurved with their anterior end below the foramen jugulare anterius, and contacts the opisthotic (figure 8*a*). This contact is often absent in turtles (including *Plesiochelys planiceps*: OUMNH J1582), so that the foramen jugulare anterius is not completely encased in bone. Posteriorly, the footplate of the exoccipital of *Sandownia harrisi* forms a hemispheric surface that forms part of the condylus occipitalis. Between the condylus occipitalis and the pterygoid contact, the occipital surface of the exoccipital forms a deeply recessed fossa ventral to the hypoglossal foramina. This fossa, which is also present in *Solnhofia parsonsi* (TM 4023) and *Plesiochelys planiceps* (OUMNH J1582), extends onto the basioccipital, where is it ventrally bound by short ridges extending from the condylus occipitalis to the tuberculum basioccipitale. These ridges form the ventral margin of the fossa that extends onto the basioccipital (see basioccipital). While these ridges and the respective fossae are also present in *Solnhofia parsonsi* (TM 4023) and *Plesiochelys planiceps* (OUMNH J1582), there is a third ridge on the basioccipital of *Sandownia harrisi* that is absent in the other two taxa. This ridge extends horizontally along the ventral margin of the basioccipital, and connects both tubercula basioccipitale. A central fossa ventral to the condylus occipitalis is formed by this ridge.

### 3.1.18. Basioccipital

The basioccipital of *Sandownia harrisi* is a roughly triangular block of bone at the posterior end of the skull midline (figure 3*d,f*). It contacts the exoccipitals, with which it forms the condylus occipitalis. The ventral surface of the basioccipital is completely covered by the pterygoids (figure 3*b*). The exposure of the basioccipital in the floor of the braincase is relatively narrow between the exoccipitals, and has a smooth surface as in *Solnhofia parsonsi* (TM 4023). A basis tuberculi basalis, as present in *Plesiochelys planiceps* (OUMNH J1582), is absent. The tubercula basioccipitale of *Sandownia harrisi* is a well developed, knob-like process at the posterolateroventral part of the bone (figure 3*f*). However, it is covered ventrally by the pterygoid, and only internally exposed medial to the carotid trough on the pterygoid (figure 6*b*; see also pterygoid). The posterior surface of the basioccipital is marked by two ridges that extend from the condylus occipitalis to the tubercula basioccipitale.

### 3.1.19. Prootic

The prootic of *Sandownia harrisi* forms the anteromedial portion of the otic capsule (figures 3*a,d*, 5, 8*a* and 9). The prootic is mediolaterally broader than anterolaterally wide and contacts the quadrate laterally, the pterygoid ventrally, the opisthotic posteriorly, the supraoccipital posterodorsally, and it has a small contact with the parabasisphenoid medioventrally.

The prootic of *Sandownia harrisi* forms the medial half of the well-developed, and deeply mediolaterally concave facet of the processus trochlearis oticum (figure 5). The anterior margin of the processus trochlearis oticum protrudes relatively far into the subtemporal fossa both in *Sandownia harrisi* and *Solnhofia parsonsi*, so that the trochlea appears very large and conceals the external trigeminal foramen completely in lateral view. The foramen stapedio-temporale is located on the dorsal surface of the otic process in *Sandownia harrisi*, and is nearly completely enclosed by the prootic, with a minor contribution by the quadrate (figure 4*a*). This is unlike *Solnhofia parsonsi* (TM 4023) or *Plesiochelys planiceps* (OUMNH J1582), in which the quadrate makes a large contribution. In *Sandownia harrisi* and *Plesiochelys planiceps* (OUMNH J1582), the stapedial artery, which exits through the foramen stapedio-temporale, leaves a deep, posteriorly recurved channel on the dorsal surface of the prootic. The contacts between the prootic and the supraoccipital and the opisthotic are unsutured, planar contacts. However, the prootic of *Sandownia harrisi* has very thin, sheet-like extensions that overlap the other two bones on the dorsal surface of the otic process.

The interior of the prootic houses the anterior part of the cavum labyrinthicum, as well as the ventral part of the anterior semicircular canal and the anterior part of the lateral semicircular canal (figure 9). The lateral semicircular canal is completely enclosed by bone of the prootic (i.e. the opisthotic does not contribute to the canal). The part of the posterior surface of the prootic that forms the anterior wall of the cavum acustico-jugulare houses a deep recess, which is positioned immediately lateral to the prootic part of the fenestra ovalis (figure 6*b*). It is possible that this space is associated with the perilymphatic recess [18]. Such a recess is also clearly developed in *Plesiochelys planiceps* (OUMNH J1582), but absent in *Solnhofia parsonsi* (TM 4023). The fossa acustico-facialis is positioned on the medial surface on the base of the ventral process of the prootic that contacts the parabasisphenoid (figure 9). A short but broad, undivided canal for the acoustic (VIII) nerve extends into the cavum labyrinthicum, whereas the facial nerve (VII) canal extends laterally through the prootic just ventral to the cavum labyrinthicum, and enters the canalis cavernosus just dorsal to the position of the foramen pro rami nervi vidiani in the pterygoid.

The prootic of *Sandownia harrisi* forms the dorsal roof of the canalis cavernosus. At its anterior end, the prootic forms a wide cavum epiptericum (figures 4 and 9*b*). This fossa houses the trigeminal ganglion and is partially concealed in lateral view by the prominent processus trochlearis oticum. The prootic is completely excluded from the formation of the external trigeminal foramen by a posteroventral process of the parietal that contacts the pterygoid along the lateral margin of the cavum epiptericum (figure 5). Prootic exclusion from the external trigeminal foramen can also be seen in *Solnhofia parsonsi* and *Plesiochelys planiceps*. The cavum epiptericum of *Sandownia harrisi* is the largest of all sandownids investigated.

### 3.1.20. Opisthotic

The opisthotic of *Sandownia harrisi* (figures 3 and 6) forms a wing-like posterolateral paroccipital process that extends from a central part that forms the posterior portion of the inner ear cavity, and a ventrally

directed processus interfenestralis. The opisthotic contacts the prootic anteriorly, the supraoccipital mediodorsally, the exoccipital posteromedioventrally and the quadrate anterolaterally.

The processus interfenestralis is a relatively delicate structure that projects ventrally (figure 6*b*). Its ventral end does not contact the pterygoid, so that there is a hiatus postlagenum. The processus interfenestralis also does not contact the prootic, so that the fenestra ovalis is ventrally open. The posteromedial margin of the processus interfenestralis bears a concave notch that forms the ventrally open fenestra perilymphatica (figure 6*b*). Medial to the fenestra perilymphatica, the opisthotic contacts the exoccipital to form the foramen jugulare anterius (figure 9*b*). The opisthotic is pierced by a short, mediolaterally directed canal for the glossopharyngeal (IX) nerve. As in other turtles, the respective foramina are clearly seen at the base of the processus interfenestralis (figure 6*b*).

The paroccipital process of *Sandownia harrisi* is dorsomedially flattened and has a ridge-like posterior margin that forms parts of the large fenestra postotica (figure 6*a*). This is different from the morphology of *Solnhofia parsonsi* (TM 4023), in which the occipital side of the opisthotic forms a broader surface. In *Plesiochelys planiceps* (OUMNH J1582), the paroccipital process is more rod-like, and has a dorsoventrally convex posterior surface.

### 3.1.21. Parabasisphenoid

The parabasisphenoid of *Sandownia harrisi* (figures 3*b–d* and 8) is unusual in several features. As shared with other sandownids, the parabasisphenoid has only a relatively small ventral exposure, and is surrounded by the pterygoids (figure 3*b*). Unlike most other turtles, the parabasisphenoid of *Sandownia harrisi* does not have a contact with the basioccipital. Instead, there is a wide transverse gap between both these bones that is superficially covered by the pterygoid (figures 4*a* and 8*a*). In *Solnhofia parsonsi* (TM 4023) and *Plesiochelys planiceps* (OUMNH J1582), the parabasisphenoid is posteroventrally expanded and covers parts of the basioccipital, and these elements share a tight contact along their common suture. The absence of a basioccipital-parabasisphenoid contact has also been described for the protostegid *Rhinochelys pulchriceps* [8], but it is possible that this is an ontogenetic artefact for that species, as both bones clearly contact each other in larger protostegid skulls, such as *Bouliachelys suteri* [21].

The lateral surface of the parabasisphenoid of *Sandownia harrisi* contacts the prootic dorsally, and the pterygoid medially. At the level of the fenestra caroticus, the pterygoid contact of the parabasisphenoid is developed as a small but distinct basipterygoid process (figure 8*c*), which is better preserved or developed on the left side.

The central part of the parabasisphenoid of *Sandownia harrisi* forms a very low cup, which is only very weakly concave dorsally (figure 8*a,d*). The clinoid processes at the anterolateral margin of the parabasisphenoid cup are extremely weakly developed bumps. The dorsum sellae is extremely low (figure 8*d*), much lower than in *Solnhofia parsonsi* (TM 4023; [19]) or *Plesiochelys planiceps* (OUMNH J1582; [22]). The anterior surface of the dorsum sellae is flat in *Sandownia harrisi*, whereas *Plesiochelys planiceps* (OUMNH J1582; [22]) has a vertical ridge spanning across this surface. *Sandownia harrisi* has very long and extremely widely spaced trabeculae (figure 9), but the medial area between them is not connected by a horizontal lamina (forming a flat rostrum basisphenoidale) as is the case in *Solnhofia parsonsi* (TM 4023; [19]) or *Plesiochelys planiceps* (OUMNH J1582; [22]). The trabeculae of *Sandownia harrisi* are furthermore dorsally slightly recurved and weakly laterally oriented, so that they project toward the sulcus cavernosus, of which they form the anterior floor. The trabeculae form parts of the foramen anterius canalis carotici palatinum (figures 7*g* and 8*c,d*). Each trabecula has a small lateral flange at its anterior tip, which contacts the parietal. Canals or foramina for the abducens (VI) nerves were not found in *Sandownia harrisi*. The abducens nerve canal usually penetrates the anterior half of the parabasisphenoid, and extends from within the parabasisphenoid cup to the dorsum sellae, where the exiting foramina are usually positioned near the base of the clinoid processes (as for instance in *Solnhofia parsonsi*). The absence of such canals in *Sandownia harrisi* is herein interpreted as a real anatomical feature, because all other nerve canals of similar size are generally observed in our CT data. Possibly, the low development of the dorsum sellae in *Sandownia harrisi* has consequentially led to the loss of a bony encasement of the abducens nerves. This interpretation is supported by a very shallow and faint groove to either side of the parabasisphenoid, that extends along the lateral margins of the anterior portion of the basisphenoid. It seems probable that the abducens nerves were placed within these grooves (figure 8).

Because the area between the trabeculae is not ossified in *Sandownia harrisi*, a true sella turcica and rostrum basisphenoidale are absent. The foramina anterius canalis carotici cerebralis, which usually

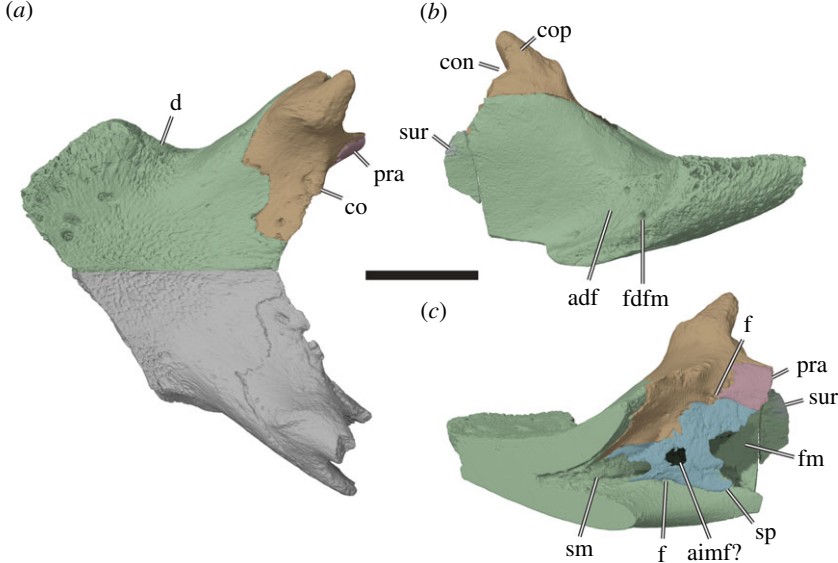

**Figure 10.** Three-dimensional renderings of the mandible of the holotype specimen of *Sandownia harrisi* (MIWG 3480). (*a*) Dorsal view. (*b*) Right lateral view. (*c*) Medial view on right mandibular ramus. adf, adductor fossa; aimf, anterior intermandibular foramen; an, angular; co, coronoid; con, coronoid notch; cop, coronoid process; d, dentary; f, unidentified foramen; fdfm, foramen dentofaciale majus; fm, fossa Meckelii; pra, prearticular; sm, sulcus Meckelii; sp, splenial; sur, surangular. Note that right and left dentaries are fused in *Sandownia harrisi* and that the horizontal cut through the dentaries has been artificially included to facilitate illustration of the medial side. Scale bar, 20 mm.

enter the sella turcica ventral to the dorsum sellae are ventrally closed by the pterygoid in *Sandownia harrisi* (figure 7*g*). In *Solnhofia parsonsi* (TM 4023; [19]), *Plesiochelys planiceps* (OUMNH J1582; [22]), as well as other thalassochelydians, the foramina anterius canalis carotici cerebralis are completely contained within the parabasisphenoid, and placed relatively closely to one another within the sella turcica.

### 3.1.22. Stapes

The stapes of *Sandownia harrisi* is a relatively narrow rod (figure 6*a,c*), like in all turtles with a tympanic middle ear (e.g. [26]). However, whereas the stapes is relatively straight and only very gently ventrally bowed in most turtles, including *Solnhofia parsonsi* (TM 4023; [19]) and *Plesiochelys planiceps* (OUMNH J1582; [22]), the stapes of *Sandownia harrisi* has a more sinuous shape: The lateral two thirds of the stapes are relatively strongly bowed posteroventrally (figure 6*c*). The medial third then curves strongly medially, extending toward the centre of the fenestra ovalis. The medial end of the stapes of *Sandownia harrisi* is not expanded to a footplate, unlike in most other turtles. However, it is likely that the stapedial footplate of *Sandownia harrisi* is simply less ossified than in most turtles, as the stapedial footplate is essential for effective sound energy transmission from the tympanum to the inner ear.

### 3.1.23. Dentary

The dentaries of MIWG 3480 are incompletely preserved. The dentaries are completely fused with one another in *Sandownia harrisi*, and the model produced for the right dentary has been cut from the left dentary along the midline of the triturating surface to allow visualization of a medial view (figure 10). The dentary of *Sandownia harrisi* contacts the coronoid dorsally, the surangular posteriorly and the splenial medially. Contacts with the prearticular, angular and articular are not preserved.

The fused dentaries of *Sandownia harrisi* form a large, flat triturating surface that is both anteroposteriorly shorter and mediolaterally broader than in *Solnhofia parsonsi* (TM 4023). Although the posterior end of the mandible is not preserved in *Sandownia harrisi*, it seems that the triturating surface would have extended over nearly half the length of the mandible. A median keel on the dorsal surface of the triturating surface is absent, as is labial ridge (i.e. there is no dorsally projecting ridge). Instead, the lateral margin of the triturating surface is highly vascularized (figure 10*b*). The triturating surface is mediolaterally widest at mid-length level of the preserved dentary length (figure 10*a*). The anterolateral margins converge towards the midline making the fused dentaries

constricted posterior to the widest extent of the triturating surface. In this area, the dentary forms a posteroventrally and laterally sloping surface that lies lateral to the dorsal process of the dentary that reaches the coronoid (figure 10b), as also seen in *Brachyopsemys tingitana* [5] and *Leyvachelys cipadi* [4]. As the lateral surface of the dorsal process of the dentary is not recessed to a fossa in *Sa. harrisi*, it seems likely that the posteroventrolaterally sloping surface mentioned above served as an attachment site for jaw musculature at the anterior end of the adductor fossa that extends over most of the posterior part of the lateral surface of the dentary. The foramen dentofaciale majus is positioned at the base of this sloping surface, but it is a relatively small foramen (figure 10b).

On the medial side of the mandible, the fused dentaries form an anteromedial pocket, the sulcus cartilaginis Meckelii, that extends deeply anteroventrally to the level of the triturating surface (figure 10c). Posterolaterally, this sulcus is connected with the fossa Meckelii via a broad canal formed by the dentary and splenial. Posterior to the connection of the sulcus cartilaginis Meckelii with the fossa Meckelii, the dentary forms a posteriorly exposed surface that forms the anterior wall of the fossa Meckelii. Within this surface is a large foramen alveolare inferius. As the foramen dentofaciale majus is much smaller than the foramen alveolare inferius, most of the anterior mandibular blood supply was likely achieved by the foramen alveolare inferius. In *Solnhofia parsonsi* (TM 4023), the foramen alveolare inferius is positioned in the medial surface of the dentary, and a posteriorly facing surface that bounds the anterior limit of the fossa Meckelii is absent.

### 3.1.24. Angular

Meylan *et al.* [2] tentatively identified an angular for *Sandownia harrisi*, but noted that the respective bone could also represent a splenial. Here, we interpret the respective bone as a splenial (see below) and conclude that the angular is not preserved in MIWG 3480.

### 3.1.25. Prearticular

Only a short part of the prearticular is present along the posteromedial surface of the coronoid (figure 10c).

### 3.1.26. Splenial

MIWG 3480 has a wedge-shaped plate of bone at the medial side of the fossa Meckelii, which we herein interpret to be the splenial (contra [2]) (figure 10c). The splenial of *Sandownia harrisi* becomes dorsoventrally lower anteriorly, and contacts the ventral surface of the anteroventral process of the coronoid, a short portion of the dentary in the medial side of the floor of the fossa Meckelii, and the prearticular posterodorsally. These features are consistent with our interpretation of the identity of the bone. Comparisons with *Plesiochelys* spp. (OUMNH J1582; [22]), *Nuesticemys neuquina* [38], and *Solnhofia parsonsi* (TM 4023) also show that these taxa have elements identified as splenials that are generally very similar in size and shape. Additionally, well preserved mandibles for several species of *Plesiochelys* and *Nuesticemys neuquina* show that the angular is usually further posteroventrally positioned than the splenial, and does not extend this far anteriorly in related taxa [22,38]. The angular, on the other hand, usually forms the ventral margin of the internal side of the mandibular ramus, whereas the parts in question in *Sandownia harrisi* are positioned more dorsally, and form the ventral coverage of the fossa Meckelii.

The splenial of *Sandownia harrisi* has a large foramen near its anterior end, which connects the fossa Meckelii with exterior of the mandible. The identity of this foramen is not clear (possibly anterior intermandibular foramen *sensu* [8]), but a much smaller foramen in a similar position has also been figured for *Plesiochelys etalloni* [22], whereas no foramen is present in *Solnhofia parsonsi* (TM 4023).

### 3.1.27. Surangular

Only a small broken piece of the surangular is preserved in articulation with the posterior part of the right dentary (figure 10b,c).

### 3.1.28. Coronoid

The coronoids are preserved on both sides of the mandible of MIWG 3480, but the left one is missing the tip of the coronoid process (figure 10). The coronoid of *Sandownia harrisi* has an extensive contact with

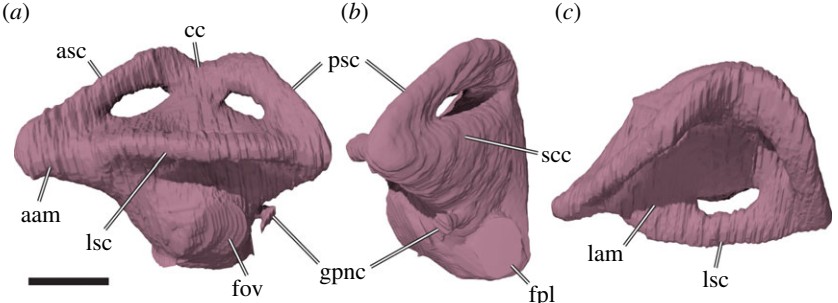

**Figure 11.** Three-dimensional renderings of the left endosseous labyrinth of *Sandownia harrisi* (MIWG 3480). (*a*) Lateral view. (*b*) Posterior view. (*c*) Dorsal view. aam, anterior ampulla; asc, anterior semicircular canal; cc, common crus; fov, fenestra ovalis; fpl, foramen perilymphatica; gpnc, glossopharyngeal nerve canal; lam, lateral ampulla; lsc, lateral semicircular canal; psc, posterior semicircular canal; scc, secondary common crus. Scale bar, 3 mm.

the dentary along its anterior and lateral surface, and further contacts the surangular posterodorsally on the medial side, the prearticular posterodorsomedially, and the splenial ventromedially.

The coronoid forms an arch that roofs the fossa Meckelii with most of its ventral surface. On the dorsal surface, the coronoid forms a long and pointed, cone-shaped coronoid process (figure 10). This morphology is very similar to that of *Solnhofia parsonsi* (TM 4023). However, the coronoid process of *Sandownia harrisi* has a small but distinct ridge on the anterolateral side of the process (figure 10*b*), which is absent in *So. parsonsi.* The coronoid process of *Sandownia harrisi* is gently recurved, creating the coronoid notch that is also present in thalassochelydians, including *Solnhofia parsonsi* [6]. Anterior to the coronoid process, the dorsal surface of the coronoid of *Sa. harrisi* forms a ventrally sloping, flat plane that contributes to the triturating surface (*contra* [2]) (figure 10*a*), that is otherwise formed by the dentary. The medial margin of the coronoid is developed as a sharp-edged rim, but it does not form an elevated lingual ridge. The coronoid contacts the splenial and surangular on the medial side of the mandible via an anteroventrally directed process (figure 10*c*). This process is very long in *Sandownia harrisi*, and extends into the sulcus Meckelii. At the contact with the splenial and prearticular, there is a small foramen that connects the medial surface of the mandible with the fossa Meckelii. This unnamed foramen is entirely located within in the coronoid in *Solnhofia parsonsi* (TM 4023) and *Plesiochelys etalloni* [22].

### 3.1.29. Endosseous labyrinth

The vertical semicircular canals in *Sandownia harrisi* (MIWG 3480) are unusually straight, and most of the curvature of the canal is accommodated near the common crus and the ampullae (figure 11*a*). The common crus is positioned ventrally below the dorsalmost extent of the anterior or posterior semicircular canal, thereby forming an 'M'-shape. The anterior and posterior semicircular canals (ASC and PSC, respectively) intersect one another at an angle of about 73° (measured on a two-dimensional image of the labyrinth in dorsal view; figure 11*c*). This is similar to the divergence angle reported for the protostegid *Rhinochelys pulchriceps* [18], but much lower than for most turtles (S.W.E., personal observation, 2019). Divergence angles have rarely been recorded for turtles, but the available literature and estimations from figures suggest that the angles range from low values as reported here, to values close to 90°, as for the plesiochelyid *Plesiochelys etalloni* [39], and to extremely wide angles, as shown for the early stem turtle *Proganochelys quenstedti* [40]. Notably, the endosseous labyrinth of *Sandownia harrisi* is markedly asymmetrical: the ASC is much longer than the PSC, and the anterior ampulla is positioned far anterior to the lateral ampulla (figure 11*a,c*). The lateral semicircular canal (LSC) is more strongly curved in its central part than either of the vertical semicircular canals. The plane of the LSC is roughly parallel to the skull roof of *Sandownia harrisi*. Posteriorly, the LSC is confluent with the ventral part of the PSC via a wide secondary common crus (figure 11*b*), as in all turtles [18]. The ASC and LSC have oval cross-sections, whereas the PSC is more circular and also somewhat thicker than the two other canals. Thickened semicircular canals have been observed in pelagic sauropterygians and chelonioids [41] and could represent an adaptation for deep diving, as the thickest semicircular canals are observed in particularly deep-diving species, such as *Dermochelys coriacea* [18]. *Sandownia harrisi*, for which a marine lifestyle is inferred, has moderately thick semicircular canals, possibly indicating it was not a deep diver.

# 4. Systematic palaeontology

Testudinata Klein, 1760 *sensu* Joyce *et al.*, 2004
Sandownidae Tong & Meylan, 2013 *sensu* Evers & Benson, 2019
*Sandownia* Meylan *et al.*, 2000

Type species: *Sandownia harrisi* Meylan *et al.*, 2000 [2].

Differential diagnosis: *Sandownia harrisi* shows numerous features that have been recovered to be synapomorphic for Angolachelonia (*sensu* [6], i.e. thalassochelydians + sandownids). These are: foramen stapedio-temporale concealed in dorsal view by a relatively extensive skull roof; presence of an infolding ridge on the posterior surface of the quadrate; articular processes of the quadrate positioned level with or posterior to the foramen magnum; presence of a very long interpterygoid contact that exceeds the length of the parabasisphenoid; pterygoid contact with the medial edge of the mandibular condyle; presence of a coronoid notch; presence of a foramen at the anterior end of the medial process of the coronoid, which leads into the fossa Meckelii; and dorsally tall and pointed coronoid process.

*Sandownia harrisi*, other sandownids, and *Solnhofia parsonsi* share an anteroposteriorly elongate vomer with a deep median trough that extends along the entire ventral surface. *Sandownia harrisi* and *Solnhofia parsonsi* share a contribution of the quadratojugal to the processus trochlearis oticum. In *Brachyopsemys tingitana*, the processus trochlearis oticum is described to be only formed by the quadrate and prootic [5], whereas in *Angolachelys mbaxi*, the provided line drawings indicate that a quadratojugal contribution to the process is conceivably present [3]. *Solnhofia parsonsi* and sandownids also share an extreme reduction of the foramen palatinum posterius, which seems to be entirely closed in *A. mbaxi*. *Sandownia harrisi*, *So. parsonsi* and *B. tingitana* share a coronoid contribution to the triturating surface of the mandible. This feature could not be investigated for other sandownids.

*Sandownia harrisi* is the essential member in the phylogenetic definition of Sandownidae [6]. Sandownid synapomorphies according to that study are the presence of only weak cheek emargination formed by the jugal and quadratojugal; the presence of a vomer–palatine contact anterior to the internal naris; a contribution of the vomer to the triturating surface; the presence of a posteromedial wing of the pterygoid that partially or completely covers the basisphenoid; and a deep dorsum sellae. Furthermore, *Sa. harrisi*, *Angolachelys mbaxi* and *Leyvachelys cipadi*, but not *Brachyopsemys tingitana*, share slightly protruding premaxillary parts of the labial margin of the skull, which is best seen in ventral view. *Sandownia harrisi*, *A. mbaxi* and *L. cipadi* also share a closed incisura columella auris, whereas this structure is posteriorly open in *B. tingitana*.

*Sandownia harrisi* can be distinguished from other sandownids by an undivided external naris that is slightly divided by an upturned ridge on the premaxillae resulting in an inverted heart shape of the naris; the absence of an epipterygoid, otherwise only reported for *Leyvachelys cipadi* [17]; a relatively weakly developed infolded ridge of the quadrate; a mediolaterally broadly exposed vomer; the presence of extremely small, fused nasals; and a posteromedial process of the jugal that extends further posteriorly along the medial margin of the subtemporal fenestra than in other sandownids. *Sandownia harrisi* furthermore shows several anatomical features that are unique to the species. However, it should be noted that other sandownids have not been characterized to the same level of detail. Consequentially, some of the following features could turn out to be sandownid synapomorphies, rather than *Sandownia harrisi* autapomorphies: presence of a large foramen in the sutural area between pterygoid and palatine, connecting the narial passage with the interorbital fenestra; a relatively strongly sinusoidal stapes without ossified stapedial footplate; a dorsally raised and anteroposteriorly oriented ridge on the pterygoid that defines a fossa within the cavum acustico-jugulare; a dorsally open trough for the internal carotid artery that extends medially to the cavum acustico-jugulare through the pterygoid; a foramen supramaxillare that is positioned within the jugal, rather than the maxilla; incomplete ossification of the abducens nerve canals in the parabasisphenoid; presence of a fenestra caroticus; presence of small basipterygoid processes in the parabasisphenoid; presence of a small parabasisphenoid-parietal contact at the anterolateral corner of the parabasisphenoid trabeculae; incomplete parabasisphenoid encasement of the cerebral artery canal, which is instead floored by the pterygoid; presence of a mandibular artery foramen within the crista pterygoidei of the pterygoid; a pterygoid contribution to the processus trochlearis oticum; the absence of epipterygoid processes on the quadrate.

Holotype: MIWG 3480, a nearly complete cranium and mandible.

Type locality and horizon: Foreshore at Atherfield Point, south coast, Isle of Wight, England. Lower Lobster Bed, Lower Greensand, Early Cretaceous (*Deshayites forbesi* Zone) [2].

Referred material: No material has so far been referred to *Sandownia harrisi*.

# 5. Discussion

## 5.1. Systematics and relationships of Sandownia harrisi

All phylogenetic analyses that include more than one sandownid find a monophyletic Sandownidae comprised of *Sandownia harrisi*, *Brachyopsemys tingitana*, *Leyvachelys cipadi* and *Angolachelys mbaxi* [3–6,8,42]. However, the phylogenetic position of Sandownidae as a clade, and its relationships with other turtle taxa, particularly *Solnhofia parsonsi*, have varied. The latest global phylogenetic studies published [6,8] found sandownids to be the sister group of thalassochelydians (*sensu* [43]), which are a group of Jurassic marine turtles from Europe and South America that include plesiochelyids, thalassemydids and eurysternids [28]. To this group, the name Angolachelonia was applied, following Mateus *et al.* [3], who defined this name as applying to sandownids + *Solnhofia parsonsi*, which was found as a thalassochelydian in [6]. Alternatively, sandownids were found to be stem-group chelonioids in some studies [4,42]. However, studies finding the latter placement did not include information that was published in a series of papers around the same time or later, which helped clarify the anatomy of at least plesiochelyids within thalassochelydians (e.g. [9,23,29,38,43–48]). These papers established some anatomical features thought to be unique to thalassochelydians, such as the infolding ridge on the quadrate [9], that have since also been observed in sandownids as well [6]. Therefore, the monophyly of sandownids with thalassochelydians seems to be well supported (see also [6]).

In this paper, we find additional support for the monophyly of Sandownidae by providing a list of features that are only observed in these turtles. Additionally, we document many features that are very similar between *Solnhofia parsonsi* and sandownids. *Solnhofia parsonsi* was found to be a thalassochelydian by Evers & Benson [6] and Evers *et al.* [8]. Although this relationship was relatively well supported in those studies, we suspect that a revised phylogenetic analysis incorporating our novel observations could move *So. parsonsi* outside of Thalassochelydia and instead place it as the sister to, or a member of, the sandownid lineage. A phylogenetic analysis necessary to test this assumption is beyond the scope of this paper, which is intended to provide an updated anatomical record of the holotype specimen of *Sandownia harrisi*. However, we are preparing such an analysis for another study.

## 5.2. Morphological innovation and early evolution of sandownids

Our new observations document a long list of features that are either autapomorphic to individual sandownid taxa, or otherwise shared among sandownids. Besides numerous unusual features of the neuroanatomical and arterial circulation system seen in *Sandownia harrisi* (e.g. unossified 'canal' for the abducens nerve; shared parabasisphenoid-pterygoid canal for the cerebral artery; presence of a mandibular artery foramen; jugal position of the foramen supramaxillare; reduction of the foramen palatinum posterius; extreme asymmetry of the endosseous labyrinth), sandownids show several cranial and mandibular modifications that are related to the formation of a secondary palate (e.g. coronoid contribution to the mandibular, and vomer contribution to the cranial triturating surfaces; modifications to the vomer, jugal, pterygoid, palatine and maxilla).

Modifications that are related to the secondary palate structure are also present in *Solnhofia parsonsi*. Particularly striking is the anteroposteriorly elongate and ventrally troughed vomer. Whereas the oldest sandownid is *Leyvachelys cipadi* from the Late Barremian or Aptian of Colombia [4], *Solnhofia parsonsi* is from the Kimmerdigian–Tithonian of Germany and Switzerland [19,49,50], and has consequentially often been hypothesized to be closely related to other secondarily marine turtles from the Late Jurassic of central Europe (i.e. thalassochelydians; e.g. [43]). However, many features of *So. parsonsi* are in closer correspondence to sandownid features than to thalassochelydian features. For instance, a secondary palate is not known for thalassochelydians outside of *So. parsonsi* [43]. Although features related to ecological adaptations, such as modifications of the palate to facilitate a durophagous diet, can evolve independently, the details of the palatal structure of *So. parsonsi* strongly suggest a close relationship with sandownids. Other features that identify *So. parsonsi* as a thalassochelydian, such as the infolding ridge on the quadrate, are now known to also be present in sandownids. *Solnhofia parsonsi* is therefore feasibly the oldest representative of the sandownid lineage, indicating that sandownids might have originated during the Late Jurassic in central Europe, and possibly evolved from thalassochelydians that were widespread in Europe around that time. The high number of morphological innovations observed for sandownids, or individual sandownid taxa, indicate relatively high rates of morphological evolution for the group, at least early in its history. Unlike thalassochelydians, such as

plesiochelyids, sandownids survived the Cretaceous/Paleogene mass extinction, as evident from their Cenozoic fossil record (e.g. [5]). Although this is speculative, it is possible that the survival of sandownids is linked to dietary specializations, as other surviving marine groups (i.e. bothremydid pleurodires and chelonioid sea turtles) also frequently show extensive secondary palates indicative of durophagous diets. We similarly note that several freshwater aquatic turtles with extensive secondary palates and durophagous diets (e.g. the baenid *Palatobaena* and plastomenid trionychids [51–53]) also survived this extinction event.

## 6. Conclusion

We provide an updated anatomical study of the holotype specimen of *Sandownia harrisi*, based on high-resolution µCT scans and derivative models. *Sandownia harrisi*, as well as other sandownids, is characterized by a large number of unusual morphological features, many of which are concentrated in neuroanatomical or circulation systems, or can be related to the evolution of a secondary palate. The great number of morphological innovations indicate high rates of morphological evolution during the early evolution of the group, and durophagy as indicated by the palatal structure of sandownids could potentially provide an explanation for sandownid survival of the Cretaceous/Paleogene mass extinction. *Solnhofia parsonsi*, a Late Jurassic turtle from central Europe, shows a large number of similarities to *Sandownia harrisi* and other related forms, and is hypothesized to represent the earliest representative of the sandownid lineage. Sandownid–thalassochelydian relationships, as well as the similarities between *Solnhofia parsonsi* and sandownids, indicate that the group originated in Europe during the Late Jurassic.

Ethics. Our work involved no live animals and we were not required to complete an ethical assessment prior to conducting our research.

Data accessibility. The CT slice data, scanning parameter information and model data used for the holotype of *Sandownia harrisi* MIWG 3480 have been published previously [16], and are available for download at MorphoSource: http://www.morphosource.org/Detail/ProjectDetail/Show/project_id/462. Scans and models for *Solnhofia parsonsi* (TM 4023) and *Plesiochelys planiceps* (OUMNH J1582) are also downloadable from the same source.

Authors' contributions. S.W.E. and W.G.J. conceived the project. S.W.E. undertook the CT scans, segmented the data and made the figures. S.W.E. and W.G.J. interpreted the data and co-wrote the MS.

Competing interests. We declare we have no competing interests.

Funding. S.W.E. was supported by a NERC studentship on the DTP Environmental Research (NE/L0021612/1). S.W.E. and W.G.J. were both funded by SNF 200021_178780/1.

Acknowledgements. We would like to thank Alex Peaker from MIWG for facilitating the scan on which this work is based upon by organizing the specimen transport, and allowing for a spontaneous loan and transfer to another facility when the CT scanner in London broke. We are also extremely thankful to Paul Barrett at the NHM London and Matt Friedman, then University of Oxford for helping to resolve issues around scanning the material in Bristol. We are very thankful to the Bristol Palaeontology CT facility and particularly Tom Davies for his excellent skills with the scanner. Thanks to Paul Barrett for providing additional photos of *Sandownia*. S.W.E. would like to thank Roger Benson (Oxford) for continued support and access to the OxfordPalaeoLab. Thanks also to Anne Schulp (Teylers Museum), Hilary Ketchum (OUMNH), Jérémy Anquetin (Jurassica) and Loïc Costeur (MNB) for access to specimens and/or facilitating CT scans of comparative material. An earlier version of the manuscript was improved by comments by Peter Meylan and another, anonymous reviewer. We thank them both for their effort and suggestions.

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
