## [Reviewer comments · Royal Society Open Science]

Review History

RSOS-191936.R0 (Original submission)

Review form: Reviewer 1 (Tomasz Szczygielski)

Is the manuscript scientifically sound in its present form?

Yes

Are the interpretations and conclusions justified by the results?

Yes

Is the language acceptable?

Yes

Do you have any ethical concerns with this paper?

No

Have you any concerns about statistical analyses in this paper?

No

Recommendation?

Accept with minor revision (please list in comments)

Comments to the Author(s)

The manuscript is concerned with skull morphology of *Sandownia harrisi*, an Early Cretaceous turtle. This is a very solid contribution. The descriptions are detailed, the comparative work is well done and the interpretations and conclusions are straightforward and justified. Identification of numerous similarities with *Solnhofia parsonsi* from the Late Jurassic is a very curious thread and consideration of these characters in a phylogenetic analysis would be very interesting. This is, unfortunately, not included in this manuscript, but the Authors clearly state that it will be done in a future work. I have no concerns with the methodology and execution of this study. Some minor corrections (formatting style oversights, mistaken bone names in some paragraphs, missing explanations of abbreviations in figure caption, etc.) and suggestions are provided in the attached PDF (Appendix A).

The figures are pivotal in a work such as this one, and fortunately they are overall great. It would be even greater, however, if the skull was also portrayed cut sagittally (right through the middle, and not slightly sideways, along the sutures, as in Fig. 3D) and horizontally, the latter in dorsal and ventral views. These three additional panels would make the descriptions easier to follow and comprehend, because currently some of the morphologies (e.g., the crista cranii, presence of the squamosal, dorsal aspect of the occiput, etc.) are not readily visible. With the 3D model at hand, preparation of these should be pretty easy and straightforward. I also think that Fig. 3 would benefit if the snapshots of the skull were captured using the orthographic rather than oblique view, limiting the distortion caused by perspective. Currently the perspective in this figure is exaggerated even compared to actual photographs (Fig. 1), enlarging the elements "closer to the camera" and causing them to obscure the structures farther away. This is alright for the remaining figures, as it gives a greater depth impression, but here I think it would be better to eliminate this effect to show more and keep the proportions between the elements constant in all views. Finally, the Authors decided to color the bones only on one half of the skull.

Unfortunately, on that side a significant part of the parietal and postorbital is missing, and the uncolored half of the skull does not present the sutures clear enough to evaluate the posterior layout of these bones. The photographs in Fig. 1 are not too helpful in that respect, either. Part of the problem may be the resolution of the figures in the review PDF, but I think that the lightning of the 3D model is also at fault. Perhaps a better shader (e.g., Radiance Scaling) would help to visualize the edges of the bones, otherwise the sutures should be indicated by lines in this figure. I am aware that the model of the skull is readily available online for examination and processing, but still I think that the paper and its figures should be self-sufficient and the figures may be labeled, so the Authors can unambiguously identify the structures they are describing.

The generic name of *Solnhofia parsonsi* is variably abbreviated throughout the manuscript as either *S. parsonsi* or *So. parsonsi*. The second option is preferred, following the ICZN suggestions (Chapter 7, Article 25, Recommendation 25A), to avoid confusion, and this should be homogenized. Furthermore, *Sandownia harrisi* should be abbreviated as *Sa. harrisi* for the same reason.

Review form: Reviewer 2**Is the manuscript scientifically sound in its present form?**

Yes

Are the interpretations and conclusions justified by the results?

Yes

Is the language acceptable?

Yes

Do you have any ethical concerns with this paper?

No

Have you any concerns about statistical analyses in this paper?

No

Recommendation?

Accept with minor revision (please list in comments)

Comments to the Author(s)

The paper is generally well-written and very well illustrated. I have made minor comments in track changes throughout.

I would discourage the authors from using the terms upper and lower temporal emargination but instead use temporal emargination for the former and cheek emargination for the latter. Upper and lower temporal emargination are going to get confused with upper and lower temporal openings in diapsids which these are not related to.

There is a significant amount of text dealing with the squamosal but none of the figures indicate this element.

The authors often use an adverb where they need an adjective. I have marked these when noted but have probably missed many.

Gaffney made it a point to describe paired structures in the singular tense to avoid the confusion that there might be more than one of them per side, I think this is a useful convention and it is not always followed in this paper.

Out of necessity, the morphological descriptions are long and detailed. I would encourage the authors to delete any details that are not likely to be phylogenetically valuable and to make sure the descriptions are concise as possible. They should do one more round of work on all the descriptions to make them as tight as possible and check that all adverbs are used correctly and that paired features are treated consistently in the singular.

Please see additional comments (Appendix B) and suggestions in marked up document.

Decision letter (RSOS-191936.R0)

20-Jan-2020

Dear Dr Evers,

On behalf of the Editors, I am pleased to inform you that your Manuscript RSOS-191936 entitled "A re-descripton of *Sandownia harrisi* (Testudinata: Sandownidae) from the Aptian of the Isle of Wight based on computed tomography scans" has been accepted for publication in Royal Society Open Science subject to minor revision in accordance with the referee suggestions. Please find the referees' comments at the end of this email.

The reviewers and handling editors have recommended publication, but also suggest some minor revisions to your manuscript. Therefore, I invite you to respond to the comments and revise your manuscript.

- Ethics statement

If your study uses humans or animals please include details of the ethical approval received, including the name of the committee that granted approval. For human studies please also detail

whether informed consent was obtained. For field studies on animals please include details of all permissions, licences and/or approvals granted to carry out the fieldwork.

- Data accessibility

If you wish to submit your supporting data or code to Dryad (<http://datadryad.org/>), or modify your current submission to dryad, please use the following link:
<http://datadryad.org/submit?journalID=RSOS&manu=RSOS-191936>

- Competing interests

- Authors' contributions

- Acknowledgements

- Funding statement

Because the schedule for publication is very tight, it is a condition of publication that you submit the revised version of your manuscript before 29-Jan-2020. Please note that the revision deadline will expire at 00.00am on this date. If you do not think you will be able to meet this date please let me know immediately.

If your manuscript is newly submitted and subsequently accepted for publication, you will be asked to pay the article processing charge, unless you request a waiver and this is approved by Royal Society Publishing. You can find out more about the charges at <https://royalsocietypublishing.org/rsos/charges>. Should you have any queries, please contact opscience@royalsociety.org.

Once again, thank you for submitting your manuscript to Royal Society Open Science and I look

forward to receiving your revision. If you have any questions at all, please do not hesitate to get in touch.

on behalf of Professor Marcelo Sanchez (Associate Editor) and Kevin Padian (Subject Editor)
openscience@royalsociety.org

Reviewer comments to Author:

Reviewer: 1
Comments to the Author(s)

The manuscript is concerned with skull morphology of *Sandownia harrisi*, an Early Cretaceous turtle. This is a very solid contribution. The descriptions are detailed, the comparative work is well done and the interpretations and conclusions are straightforward and justified. Identification of numerous similarities with *Solnhofia parsonsi* from the Late Jurassic is a very curious thread and consideration of these characters in a phylogenetic analysis would be very interesting. This is, unfortunately, not included in this manuscript, but the Authors clearly state that it will be done in a future work. I have no concerns with the methodology and execution of this study. Some minor corrections (formatting style oversights, mistaken bone names in some paragraphs, missing explanations of abbreviations in figure caption, etc.) and suggestions are provided in the attached PDF.

The figures are pivotal in a work such as this one, and fortunately they are overall great. It would be even greater, however, if the skull was also portrayed cut sagittally (right through the middle, and not slightly sideways, along the sutures, as in Fig. 3D) and horizontally, the latter in dorsal and ventral views. These three additional panels would make the descriptions easier to follow and comprehend, because currently some of the morphologies (e.g., the crista cranii, presence of the squamosal, dorsal aspect of the occiput, etc.) are not readily visible. With the 3D model at hand, preparation of these should be pretty easy and straightforward. I also think that Fig. 3 would benefit if the snapshots of the skull were captured using the orthographic rather than oblique view, limiting the distortion caused by perspective. Currently the perspective in this figure is exaggerated even compared to actual photographs (Fig. 1), enlarging the elements “closer to the camera” and causing them to obscure the structures farther away. This is alright for the remaining figures, as it gives a greater depth impression, but here I think it would be better to eliminate this effect to show more and keep the proportions between the elements constant in all views. Finally, the Authors decided to color the bones only on one half of the skull. Unfortunately, on that side a significant part of the parietal and postorbital is missing, and the uncolored half of the skull does not present the sutures clear enough to evaluate the posterior layout of these bones. The photographs in Fig. 1 are not too helpful in that respect, either. Part of the problem may be the resolution of the figures in the review PDF, but I think that the lightning of the 3D model is also at fault. Perhaps a better shader (e.g., Radiance Scaling) would help to visualize the edges of the bones, otherwise the sutures should be indicated by lines in this figure. I am aware that the model of the skull is readily available online for examination and processing, but still I think that the paper and its figures should be self-sufficient and the figures may be labeled, so the Authors can unambiguously identify the structures they are describing.

The generic name of *Solnhofia parsonsi* is variably abbreviated throughout the manuscript as either *S. parsonsi* or *So. parsonsi*. The second option is preferred, following the ICZN suggestions

(Chapter 7, Article 25, Recommendation 25A), to avoid confusion, and this should be homogenized. Furthermore, *Sandownia harrisi* should be abbreviated as *Sa. harrisi* for the same reason.

Reviewer: 2

Comments to the Author(s)

The paper is generally well-written and very well illustrated. I have made minor comments in track changes throughout.

I would discourage the authors from using the terms upper and lower temporal emargination but instead use temporal emargination for the former and cheek emargination for the latter. Upper and lower temporal emargination are going to get confused with upper and lower temporal openings in diapsids which these are not related to.

There is a significant amount of text dealing with the squamosal but none of the figures indicate this element.

The authors often use an adverb where they need an adjective. I have marked these when noted but have probably missed many.

Gaffney made it a point to describe paired structures in the singular tense to avoid the confusion that there might be more than one of them per side, I think this is a useful convention and it is not always followed in this paper.

Out of necessity, the morphological descriptions are long and detailed. I would encourage the authors to delete any details that are not likely to be phylogenetically valuable and to make sure the descriptions are concise as possible. They should do one more round of work on all the descriptions to make them as tight as possible and check that all adverbs are used correctly and that paired features are treated consistently in the singular.

Please see additional comments and suggestions in marked up document.

Author's Response to Decision Letter for (RSOS-191936.R0)

See Appendix C.

Decision letter (RSOS-191936.R1)

27-Jan-2020

Dear Dr Evers,

It is a pleasure to accept your manuscript entitled "A re-description of *Sandownia harrisi* (Testudinata: Sandownidae) from the Aptian of the Isle of Wight based on computed tomography scans" in its current form for publication in Royal Society Open Science. The comments of the reviewer(s) who reviewed your manuscript are included at the foot of this letter.

Please ensure that you send to the editorial office an editable version of your accepted

manuscript, and individual files for each figure and table included in your manuscript. You can send these in a zip folder if more convenient. Failure to provide these files may delay the processing of your proof. You may disregard this request if you have already provided these files to the editorial office.

on behalf of Professor Marcelo Sanchez (Associate Editor) and Kevin Padian (Subject Editor)
openscience@royalsociety.org

Associate Editor Comments to Author (Professor Marcelo Sanchez):

Congratulations on the scholarly study. There is a typo in line 15 of page 24, please check that at the proof stage.

Appendix A**ROYAL SOCIETY
OPEN SCIENCE****A re-description of *Sandownia harrisi* (Testudinata: Sandownidae) from the Aptian of the Isle of Wight based on computed tomography scans**

Journal:	Royal Society Open Science
Manuscript ID	RSOS-191936
Article Type:	Research
Date Submitted by the Author:	04-Nov-2019
Complete List of Authors:	Evers, Serjoscha; University of Fribourg, Geosciences Joyce, Walter; University of Fribourg, Geosciences
Subject:	palaeontology < BIOLOGY, evolution < BIOLOGY
Keywords:	Sandownidae, Thalassocheyleydia, cranial anatomy, labyrinth, turtles, evolution
Subject Category:	Biology (whole organism)

Author-supplied statements

Relevant information will appear here if provided.

Ethics

Does your article include research that required ethical approval or permits?:

This article does not present research with ethical considerations

Statement (if applicable):

CUST_IF_YES_ETHICS :No data available.

Data

It is a condition of publication that data, code and materials supporting your paper are made publicly available. Does your paper present new data?:

Yes

Statement (if applicable):

The CT slice data, scanning parameter information, and model data used for the holotype of *Sandownia harrisi* MIWG 3480 have been published previously [16], and are available for download at MorphoSource: http://www.morphosource.org/Detail/ProjectDetail/Show/project_id/462. Scans and models for *Solnhofia parsonsi* (TM 4023) and *Plesiochelys planiceps* (OUMNH J1582) are also downloadable from the same source.

Conflict of interest

I/We declare we have no competing interests

Statement (if applicable):

CUST_STATE_CONFLICT :No data available.

Authors' contributions

This paper has multiple authors and our individual contributions were as below

Statement (if applicable):

SWE and WGJ conceived the project. SWE undertook the CT scans, segmented the data, and made the figures. SWE and WGJ interpreted the data and co-wrote the MS.

A re-description of *Sandownia harrisi* (Testudinata: Sandownidae) from the Aptian of the Isle of Wight based on computed tomography scans

Serjoscha W. Evers^{1*†} and Walter G. Joyce¹

¹Departement of Geosciences, University of Fribourg, Fribourg, Switzerland

*Author for correspondence: Serjoscha W. Evers, serjoscha.ever@gmail.com

†Present address: Serjoscha W. Evers, Department of Geosciences, University of Fribourg, Chemin du Musée 4, 1700 Fribourg, Switzerland.

Keywords: Sandownidae, Thalassochelydia, cranial anatomy, labyrinth, turtles, evolution

1. Summary

Sandownidae is an enigmatic group of Cretaceous–Paleogene turtles with highly derived cranial anatomy. Although sandownid monophyly is not debated, relationships with other turtles are unclear. Sandownids have been recovered in wildly different parts of the turtle tree, for instance 
[revised manuscript text omitted]
 respective observation of Meylan et al. [2]. The parietal of *S. harrisi* dorsally fully overlaps the
6 supraoccipital with its posterior and the supraoccipital is therefore concealed in dorsal view.

The descending process of the parietal of *S. harrisi* forms an anteroposteriorly extensive sheet of bone,
that contacts the palatine anteroventrally, the pterygoid ventrally, and the prootic posteroventrally (figure 3D).
An unusual feature of *S. harrisi* is a small contact of the medial surface of the descending process with the
trabecula of the parabasisphenoid at the position of the foramen anterius canalis carotici palatinum, i. e., the
anterior exiting foramen for the palatine artery (see below). At its contact with the prootic, the descending
process of the parietal forms a posterolaterally and ventrally directed ramus, which extends ventrally underneath
the prootic part of the processus trochlearis oticum (figure 4). This ramus forms the roof of the cavum
epiptericum and excludes the prootic from contributing to the margin of the external trigeminal foramen (*sensu*
[18]), which is entirely enclosed by the parietal and the pterygoid. On the dorsal surface of the otic chamber, the
parietal forms a small contribution to the deep, ventrally concave processus trochlearis oticum. A similar
condition is present in *Solnhofia parsonsi* (TM 4023), in which the respective parietal contribution to the
processus trochlearis oticum is, however, much larger.

*Postorbital*. The postorbital of *S. harrisi* forms the posterior margin of the orbit, and contacts the frontal
anteromedially, the parietal posteromedially, the jugal ventrally, and the quadratojugal posteroventrally (figure
3A, C, F). A contact with the squamosal is apparent from the small squamosal fragment which is preserved on
ventral side of the right temporal roofing. The ventral surface of the postorbital has a low, mediolateral ridge,
which continues laterally onto the jugal and medially onto the parietal. This ridge forms the weak posterior
delimitation of the orbit. This ridge is also present, albeit more massively developed, in *Solnhofia parsonsi* (TM
4023). The inferred squamosal/parietal contact implies that the postorbital did not contribute to the upper
temporal emargination.

*Jugal*. The jugal of *Sandownia harrisi* is a large bone at the lateral surface of the skull (figure 3A–C). External
contacts are present with the maxilla anteriorly, the postorbital dorsally, and the quadratojugal posteriorly. A
contact with the squamosal is absent. The jugal only forms a small portion of the posteroventral margin of the
orbit (figure 3C). The ventral margin of the jugal extends to the ventral margin of the maxilla, as in other
sandownids but unlike in *Solnhofia parsonsi* (TM 4023; [19]), in which the jugal is positioned relatively more

dorsally and is ventrally framed by an extended posterior process of the maxilla. The ventral margin of the jugal
of *S. harrisi* is concavely curved, and forms the anterior half of a moderate lower temporal emargination (figure
3C). The lower temporal emargination of *S. harrisi* is deeper than that of other sandownids [3,4,5].

[revised manuscript text omitted]

57 **Competing Interests**

We have no competing interests.

59 **Authors' Contributions**

[revised manuscript text omitted]

* Author for correspondence: Serjoscha W. Evers, serjoscha.ever@googlemail.com

† Present address: Serjoscha W. Evers, Department of Geosciences, University of Fribourg, Chemin du Musée 4, 1700 Fribourg, Switzerland.

Keywords: Sandownidae, Thalassochelydia, cranial anatomy, labyrinth, turtles, evolution

1. Summary

The turtle family Sandownidae is an enigmatic ~~Cretaceous-Paleogene~~ group of ~~Cretaceous-Paleogene turtles~~ with highly derived cranial anatomy. Although sandownid monophyly is not debated, relationships with other turtles ~~are~~ remain unclear. Sandownids have been recovered in ~~wildly~~ significantly different parts of the turtle tree; ~~for instance~~ as stem-turtles, stem-cryptodires, and stem-chelonoid sea turtles and "trionychoids". Latest phylogenetic studies find sandownids closely related to the Late Jurassic thalassochelydians ~~and~~ as stem-turtles. Here, we provide a detailed study of the cranial and mandibular anatomy of *Sandownia harrisi* from the Aptian of the Isle of Wight, based on high resolution computed tomography scanning of the holotype. Our results confirm a high number of anatomical similarities with thalassochelydians and particularly *Solnhofia parsonsi*, which is interpreted as an early member of the sandownid lineage. Sandownids + *Solnhofia* show many cranial modifications related to the secondary palate and ~~their a~~ durophagous diet, ~~and~~ *Sandownia* is additionally highly derived in features related to its arterial circulation and neuroanatomy, including the endosseous labyrinth. Our results imply rapid morphological evolution during the early history of sandownids. Sandownids likely evolved in central Europe from thalassochelydian ancestors during the Late Jurassic. The durophagous diet of sandownids possibly facilitated their survival of the Cretaceous/Paleogene mass extinction.

Commented [m1]: Can this be more specific. Close relationship is relative. I am closely related to *Amia* relative to *Paramecium*.

2. Introduction

The Early Cretaceous (~~Lower Aptian: *Deshayites forbesi* Zone [1]~~) fossil turtle *Sandownia harrisi* is known from a single specimen from Lower Aptian: *Deshayites forbesi* Zone [1] of the Isle of Wight that was initially described by Meylan *et al.* [2]. The holotype includes a well-preserved cranium (figure 1) and a partial lower jaw (figure 2).

~~New Additional fossils-new species~~ described ~~since the description of after~~ *Sandownia harrisi* have shown that the peculiar skull morphology of this turtle is shared by ~~a number of at least three~~ other ~~fossil turtle~~ species, ~~in particular~~ *Angolachelys mbaxi* [3] from the Late Turonian of Angola, *Leyvachelys cipadi* [4] from the Late Barremian or Aptian of Colombia, and *Brachyopsemys tingitana* [5] from the Danian of Morocco, ~~and~~ Tong & Meylan [5] ~~named the group comprising these four turtles recognized the~~ Sandownidae ~~on the basis of *Sandownia*, *Angolachelys*, *Brachyopsemys* and an undescribed form from the~~ of Texas. ~~and~~ Evers & Benson [6] provided a phylogenetic definition for the name.

Sandownia harrisi was originally identified as an aberrant representative of Trionychoidea [2], a ~~now-defunct no longer recognized~~ group of turtles consisting of the clades ~~Kinosternoidea~~ *Kinosternoidea* and Trionychia [7]. All phylogenetic analyses including more than one sandownid have confirmed the monophyly of this group with regard to other turtles [2,3,4,6,8]. ~~However, the~~ phylogenetic position of sandownids within the turtle tree remains ~~under dispute unclear, however,~~ with some authors ~~finding-identifying~~ them ~~variably~~ as stem-chelonioids (e.g. [4,5,9]), ~~but others as~~ stem-cryptodires [3], stem-turtles [6], stem-pleurodires [8], trionychiids (e.g. [10,11]), protostegids [12], or among the group now termed ~~Thalassochelydia~~ (e.g. [3,6,12]).

Sandownids span at least 60 million years ~~of evolutionary history,~~ and survived the mass extinction associated with the Cretaceous/Paleogene boundary. Sandownids ~~further have had a very~~ wide geographic distribution, even in the Early Cretaceous. This can possibly be explained by their ecology, as sandownids are universally interpreted as secondarily marine ~~turtles~~, because their fossils have been recovered from shallow marine depositional environments, and because the presence of extensive secondary palates and large triturating surfaces are consistent with ~~marine feeding adaptations~~ [3,5,6]. Depending on their phylogenetic position, sandownids could either be informative about the early evolution of ~~extant a previously known group of marine sea turtles, the~~ (chelonioids), ~~or the Thalassochelydia,~~ or provide ~~independent~~ evidence for ~~an independent adaptation secondarily~~

Formatted: Font: Italic

Formatted: Font: Italic

Formatted: Font: Italic

Commented [m2]: wrong citation. should probably be G&M from Benton volume

Formatted: Highlight

Commented [m3]: if these are plesiochelyids of some authors it would be worth saying so.

Commented [m4]: durophagy is not a marine feeding adaptation. There are many freshwater turtles with elaborate palates and durophage (emydids, geoemydids, trionychiids, kinosternids, etc) .

~~to marine-environments, adaptation. The latter~~Clarification of sandownid relationships is an important ~~macroevolutionary-to the~~ field of ~~macroevolutionresearch,~~ as ecological transitions provide important insights into the tempo and mode of morphological evolution (e.g., [8,13,14,15]).

[revised manuscript text omitted]

Posterolaterally, the prefrontal contacts the postorbital, and posteriorly the parietal. The lateral margin of the prefrontal forms the posterodorsal margin of the orbit. Ventrally, the frontal extends underneath the prefrontal, and forms a low and thin crista cranii. Due to the low mediolateral width of the anterior frontal process, the crista cranii of both prefrontals jointly define a narrow, ventrally open sulcus olfactorius, which is similar to that of *Brachyopsemys tingitana* [5].

Parietal. The parietal of *Sandownia harrisi* is a large bone that forms the majority of the skull roof and large parts of the braincase via the descending process (figure 3A, C–D, F). On the skull roof, the parietal contacts the frontal anteriorly and the extremely elongate postorbital laterally. The posterior margin of the parietal is not entirely preserved in either right or left element, but preserved bits portions of the original margin on the right side indicate that the posterior temporal emargination of *S. harrisi* was slightly larger than that of *Angolachelys mbaxi* [3], *Brachyopsemys tingitana* [5], or *Leyoachelys cipadi* [4]. A minor contact with the squamosal, as present in other sandownids, is preserved on the right side of the skull, as evident in our CT scans and thus confirming the respective observation of Meylan et al. [2]. The parietal of *S. harrisi* dorsally fully overlaps the supraoccipital dorsally, thus with its posterior 
[revised manuscript text omitted]
. This unnamed foramen is not visible in the figures, but can easily be observed in the models in oblique posterior view through the internal nares. The foramen has, to our knowledge, not been observed or described for any turtle before. It is a relatively large foramen, comparable to the size of the opening of foramina associated with the carotid articulation. The foramen anterius canalis carotici palatinum (faccp), i.e. the anterior exiting foramen for the palatine artery, is positioned ~~shortly just~~ posterior to the position of the unnamed foramen. Therefore, it seems possible that the palatine artery, after entering the cavum cranii, extends anteriorly to the foramen and then send off a branch that enters the choanal tunnel to supply blood to this region of the skull. In *S. parsonsi* (~~at least~~ TM 4023), this region of the palatine-ptyergoid contact is broken, so that it is not possible to see if a similar foramen is present in this taxon.

[revised manuscript text omitted]

Commented [R29]: Is this the difference you want to point out?

Formatted: Indent: First line: 2.95 pi

associated with the perilymphatic recess [18]. Such a recess is also clearly developed in *Plesiochelys planiceps* (OUMNH J1582), but absent in *Solnhofia parsonsi* (TM 4023). The fossa acustico-facialis is positioned on the medial surface on the base of the ventral process of the prootic that contacts the parabasisphenoid (figure 8). A short but broad, undivided canal for the acoustic (VIII) nerve extends into the cavum labyrinthicum, whereas the facial nerve (VII) canal extends laterally through the prootic just ventrally to the cavum labyrinthicum, and enters the canalis cavernosus just dorsal to the position of the foramen pro rami nervi vidiani in the pterygoid.

The prootic of *Sandownia harrisi* forms the dorsal roof of the canalis cavernosus. At its anterior end, the prootic forms a wide cavum epiptericum (figure 8B). This fossa houses the trigeminal ganglion and is partially concealed in lateral view by the prominent processus trochlearis oticum. The prootic is completely excluded from the formation of the external trigeminal foramen by a posteroventral process of the parietal that contacts the pterygoid along the lateral margin of the cavum epiptericum (figure 4). Prootic exclusion from the external trigeminal foramen can also be seen in *Solnhofia parsonsi* and *Plesiochelys planiceps*. Whereas the cavum epiptericum of *Sandownia harrisi* is the largest of all sandownids investigated, a prootic exclusion from the external trigeminal foramen can also be seen in *Solnhofia parsonsi* and *Plesiochelys planiceps*.

Opisthotic. The opisthotic of *Sandownia harrisi* (figures 3 and 5) forms a wing-like posterolateral paroccipital process that extend from, a central part that forms the posterior part-portion of the inner ear cavity, and a ventrally directed processus interfenestralis. The opisthotic contacts the prootic anteriorly, the supraoccipital mediodorsally, the exoccipital posteromedioventrally, and the quadrate anterolaterally.

The processus interfenestralis is a relatively delicate structure that projects ventrally (figure 5B). Its ventral end does not contact the pterygoid, so that there is a hiatus postlagenum. The processus interfenestralis also does not contact the prootic, so that the fenestra ovalis is ventrally open. The posteromedial margin of the processus interfenestralis bears a concave notch that forms the ventrally open fenestra perilymphatica (figure 5B). Medially to the fenestra perilymphatica, the opisthotic contacts the exoccipital to form the foramen jugulare anterius (figure 8B). The opisthotic is

pierced by a short, mediolaterally directed canal for the glossopharyngeal (IX) nerve. As in other turtles, the respective foramina are clearly seen at the base of the processus interfenestralis (figure 5B).

The paroccipital process of *Sandownia harrisi* is dorsomedially flattened and has a ridge-like posterior margin that forms parts of the large fenestra postotica (figure 5A). This is different from the morphology of *Solnhofia parsonsi* (TM 4023), in which the occipital side of the opisthotic forms a broader surface. In *Plesiochelys planiceps* (OUMNH J1582), the paroccipital process is more rod-like, and has a dorsoventrally convex posterior surface.

Parabasisphenoid. The parabasisphenoid of *Sandownia harrisi* (figures 3B–D and 7) is unusual in several features. As shared with other sandownids, the parabasisphenoid has only a relatively small ventral exposure, and is surrounded by the pterygoids (figure 3B). Unlike ~~in~~ most other turtles, the parabasisphenoid of *Sandownia harrisi* does not have a contact with the basioccipital. Instead, there is a wide transverse gap between ~~both~~ these bones that is superficially covered by the pterygoid (figure 7A). In *Solnhofia parsonsi* (TM 4023) and *Plesiochelys planiceps* (OUMNH J1582), the parabasisphenoid is posteroventrally expanded and covers parts of the basioccipital, and ~~both bones have these elements share~~ a tight contact along the anterior surface of the basioccipital ~~their common suture~~. ~~A missing~~ The absence of 
[revised manuscript text omitted]

Competing Interests

We have no competing interests.

Authors' Contributions

[revised manuscript text omitted]

Appendix C

Reviewer 1

The manuscript is concerned with skull morphology of *Sandownia harrisi*, an Early Cretaceous turtle. This is a very solid contribution. The descriptions are detailed, the comparative work is well done and the interpretations and conclusions are straightforward and justified. Identification of numerous similarities with *Solnhofia parsonsi* from the Late Jurassic is a very curious thread and consideration of these characters in a phylogenetic analysis would be very interesting. This is, unfortunately, not included in this manuscript, but the Authors clearly state that it will be done in a future work. I have no concerns with the methodology and execution of this study. Some minor corrections (formatting style oversights, mistaken bone names in some paragraphs, missing explanations of abbreviations in figure caption, etc.) and suggestions are provided in the attached PDF.

Response: Additional suggestions in the marked up PDF were implemented.

The figures are pivotal in a work such as this one, and fortunately they are overall great. It would be even greater, however, if the skull was also portrayed cut sagittally (right through the middle, and not slightly sideways, along the sutures, as in Fig. 3D) and horizontally, the latter in dorsal and ventral views. These three additional panels would make the descriptions easier to follow and comprehend, because currently some of the morphologies (e.g., the crista cranii, presence of the squamosal, dorsal aspect of the occiput, etc.) are not readily visible. With the 3D model at hand, preparation of these should be pretty easy and straightforward.

Response: We have taken the following steps to address these points by reviewer 1:

1. We have modified Fig. 3D so that it now displays sagittal cuts through bones that are in a median position within the skull.
2. We have added a new figure (now figure 4) that shows the horizontally sectioned; panel A shows the ventral section of the skull in dorsal view, and panel B shows the dorsal section of the skull in ventral view. In panel A, additional features that were previously not figured are now visible (e.g. accessory foramen in palatine, dorsal aspects of occiput, parts of the right squamosal). In panel B, additional features that can be seen now are the crista cranii, and further aspects of the squamosal.

I also think that Fig. 3 would benefit if the snapshots of the skull were captured using the orthographic rather than oblique view, limiting the distortion caused by perspective. Currently the perspective in this figure is exaggerated even compared to actual photographs (Fig. 1), enlarging the elements "closer to the camera" and causing them to obscure the structures farther away. This is alright for the remaining figures, as it gives a greater depth impression, but here I think it would be better to eliminate this effect to show more and keep the proportions between the elements constant in all views.

Response: We have modified the panels in figure 3 to show more orthographic view of the cranial sides figured. The largest difference is in figure 3E (anterior view).

Finally, the Authors decided to color the bones only on one half of the skull. Unfortunately, on that side a significant part of the parietal and postorbital is missing, and the uncolored half of the skull does not present the sutures clear enough to evaluate the posterior layout of these bones. The photographs in Fig. 1 are not too helpful in that respect, either. Part of the problem may be the resolution of the figures in the review PDF, but I think that the lighting of the 3D model is also at fault. Perhaps a better shader (e.g., Radiance Scaling) would help to visualize the edges of the bones, otherwise the sutures should be indicated by lines in this figure. I am aware that the model of the skull is readily available online for examination and processing, but still I think that the paper and its figures should be self-sufficient and the figures may be labeled, so the Authors can unambiguously identify the structures they are describing.

Response: In our modified figure 3, panel A now shows a better version of the right cranial side. The model has been re-segmented and allows suture tracing for right-sided elements. The sutures were added as faint lines, and the squamosal, which is now visible, was additionally labelled. Also visible now is the posterior extent of the postorbital and parietal.

The generic name of *Solnhofia parsonsi* is variably abbreviated throughout the manuscript as either *S. parsonsi* or *So. parsonsi*. The second option is preferred, following the ICZN suggestions (Chapter 7, Article 25, Recommendation 25A), to avoid confusion, and this should be homogenized. Furthermore, *Sandownia harrisi* should be abbreviated as *Sa. harrisi* for the same reason.

Response: I changed both abbreviations according to the reviewer's recommendations.

Reviewer 2

I would discourage the authors from using the terms upper and lower temporal emargination but instead use temporal emargination for the former and cheek emargination for the latter. Upper and lower temporal emargination are going to get confused with upper and lower temporal openings in diapsids which these are not related to.

Response: Done. We have exchanged the terms as suggested by the reviewer.

There is a significant amount of text dealing with the squamosal but none of the figures indicate this element.

Response: The squamosal is now figure in Fig 3A and in the new figure (figure 4) as well. See also response to comments of reviewer 1.

The authors often use an adverb where they need an adjective. I have marked these when noted but have probably missed many.

Response: The changes indicated by the reviewer in the annotated PDF were implemented. Additionally, I have gone through all adverbs again and made some additional changes that the reviewer had not marked as incorrect.

Gaffney made it a point to describe paired structures in the singular tense to avoid the confusion that there might be more than one of them per side, I think this is a useful convention and it is not always followed in this paper.

Response: We have reviewed our text and changed instances, except in cases in which there is a specific reason to talk about the bones in plural (e.g. 'fused nasals').

Out of necessity, the morphological descriptions are long and detailed. I would encourage the authors to delete any details that are not likely to be phylogenetically valuable and to make sure the descriptions are concise as possible. They should do one more round of work on all the descriptions to make them as tight as possible and check that all adverbs are used correctly and that paired features are treated consistently in the singular.

Response: We have reviewed our descriptions and tried to use adverbs correctly and used singular whenever possible when referring to specific bones. However, we have not significantly shortened our description, for the following reasons: We believe that limiting descriptions to perceptions of whatever is currently seen as phylogenetically informative is problematic, because our concept of phylogenetic characters change over time, and additionally this would preclude autapomorphic features (which, again, are only autapomorphic as long as no second taxon with the same feature is identified). Additionally, we have specifically chosen a venue with 'enough room' to publicize the anatomy of Sandownia in detail, without writing a proper monograph.

Please see additional comments and suggestions in marked up document.

Response: Reviewer 2 has made a lot of small changes to our text, for which we are most grateful. I accepted most of these suggestions, particularly those pertaining to adverb or other language errors. A few times, the reviewer deleted several sentences from our text, and we have reviewed the respective sections and then variably accepted or declined the respective suggestions. Within the annotated MS, reviewer 1 also makes a few content-related comments, and these are listed here for clarity:

1. In the systematic palaeontology section, we originally used the authorities on the phylogenetic definitins of the clades listed. Reviewer 1 suggested to use the authorities of the classic Linnean rank-system. Now, we have combined these, by citing the original first author of the name, and added 'sensu author of phylogenetic clade definition' after. This way, original authors are cited, but it becomes clear that we are applying the names according to phylogenetic nomenclature.
2. The reviewer suggested additional citations in two places, and they are now included.

Additional changes to the MS:

We noted a labelling error in figure 10 (now figure 11) that we fixed. Accordingly, a new version of the figure was uploaded.

The addition of a new figure (figure 4) as well as additional references changed the numbering of other figures and references. These were all implemented, so that references and figure mentioning refer to the correct respective references and figures.